# Antagonistic behavior of brain networks mediated by low-frequency oscillations: electrophysiological dynamics during internal–external attention switching
Jiri Hammer [1] ✉, Michaela Kajsova [1], Adam Kalina [1], David Krysl[1], Petr Fabera[1], Martin Kudr[2], Petr Jezdik[1,3], Radek Janca [1,3], Pavel Krsek[2] & Petr Marusic [1] ✉

Antagonistic activity of brain networks likely plays a fundamental role in how the brain optimizes its performance by efficient allocation of computational resources. A prominent example involves externally/internally oriented attention tasks, implicating two anticorrelated, intrinsic brain networks: the default mode network (DMN) and the dorsal attention network (DAN). To elucidate electrophysiological underpinnings and causal interplay during attention switching, we recorded intracranial EEG (iEEG) from 25 epilepsy patients with electrode contacts localized in the DMN and DAN. We show antagonistic network dynamics of activation-related changes in high-frequency (> 50 Hz) and low-frequency (< 30 Hz) power. The temporal profile of information flow between the networks estimated by functional connectivity suggests that the activated network inhibits the other one, gating its activity by increasing the amplitude of the low-frequency oscillations. Insights about inter-network communication may have profound implications for various brain disorders in which these dynamics are compromised.

Antagonistic activations between the default mode network (DMN) and other brain regions have been proposed as a fundamental physiological mechanism for how the brain orchestrates and optimizes the allocation of resources and, ultimately, its performance[1,2]. The antagonistic behavior is characterized by a state of simultaneous activation of one network and deactivation of another, resulting in their anticorrelated activity. Anticorrelations (i.e. negative correlations) between different brain networks have been observed both spontaneously during a resting state[3,4] and during a variety of cognitive tasks[5–7]. Compromised antagonistic relationships at the network level have been observed in different psychiatric disorders[1].

A prominent example of antagonistic brain activity can be observed in tasks requiring sustained external or internal attention. In tasks demanding externally oriented attention, the DMN is deactivated, while another intrinsic neural network, the dorsal attention network (DAN), is activated[8,9]. The suppression of the DMN may be due to effective allocation of the brain's computational resources and proper information flow between the DAN and other networks indispensable for

solving the task[2]. Interestingly, a lack of DMN suppression leads to suboptimal performance, typically experienced by subjects as an intrusion of task-unrelated thoughts or a momentary lapse of attention[10–13]. Hence, the antagonism of the DMN and DAN may represent functional competition between systems for attention allocation, prioritizing either internal or external information based on the current needs and situation[1].

The DMN, initially discovered as the resting state "default" (or "task-negative") network[4,5,7,9] (for recent reviews see refs. 14–17), is a large-scale neural network distributed over the association cortex, comprising areas in the frontal (medial and anterior prefrontal cortex), temporal (lateral temporal cortex and medial temporal lobe), and parietal (posterior cingulate cortex and inferior parietal lobule) lobes. Later, the DMN was shown not only to be the task-negative network, but also to directly support internally oriented cognitive processes. Specifically, it was found to be activated during perceptual decoupling[18], when internally constructed representations or internally oriented attention beyond the immediate sensory environment was required, for example, in self-referential tasks, self-episodic memory

[1]Department of Neurology, Second Faculty of Medicine, Charles University and Motol University Hospital, Prague, Czech Republic. [2]Department of Pediatric Neurology, Second Faculty of Medicine, Charles University and Motol University Hospital, Prague, Czech Republic. [3]Department of Circuit Theory, Faculty of Electrical Engineering, Czech Technical University in Prague, Prague, Czech Republic. ✉e-mail: jiri.hammer@lfmotol.cuni.cz; Petr.Marusic@fnmotol.cz

retrieval, envisioning one's own future, making social inferences, or mind wandering[15–17]. An interesting, overarching hypothesis suggests that the DMN produces and broadcasts to other brain areas an ongoing internal narrative—a continuum of mental thoughts (or an internal speech). This internal narrative helps to define our subjective continuity and a coherent sense of self and may be temporarily suspended during periods requiring external attention[17].

The DAN is activated during tasks requiring voluntary (top-down), externally oriented attention such as visuospatial attention tasks[8,19]. Its functional role is a selection of stimuli and preparation of responses in a top-down manner, in contrast to the sensory-driven, bottom-up (e.g., salient) stimulus processing of the ventral attention system[8]. The core regions of the DAN consist of the areas in the superior parietal lobule, intraparietal sulcus and frontal eye field.

Most of the evidence for the anticorrelated, antagonistic activity of the large-scale brain networks, such as the DMN and DAN, initially came from neuroimaging studies using positron emission tomography[6,7] or functional magnetic resonance imaging (fMRI)[3,5,9,19]. Although the findings from fMRI face several interpretational challenges, such as the methodological issue of global signal subtraction[20], other studies using different signal modalities, such as single- and multi-unit activity in monkeys[21], simultaneous EEG and fMRI[22] or human intracranial EEG (iEEG), have confirmed the observations from fMRI.

The iEEG studies[23–36] (for a recent review see ref. 37) confirmed the antagonistic relationship between the DMN and DAN, both during a resting state[30] and during cognitive tasks, showing that the DMN was deactivated during tasks requiring externally oriented attention, such as visual search tasks[25] or mental arithmetic[34]. Conversely, the DMN was activated during tasks requiring internally oriented attention, such as self-episodic memory retrieval[26,27], or theory of mind tasks[35].

Although the antagonistic relationship between the DMN and DAN during tasks requiring externally or internally oriented attention seems to be well established, the neural underpinnings, temporal dynamics and directionality of their interactions during the fast transitions between the activated states have not yet been clarified. In particular, which frequency bands mediate the transitions between DMN and DAN active states? Is there a temporal order of the DMN and DAN in their activity reversal during attention switching? What are the nature, directionality and dynamics of the interactions between the DMN and DAN?

To address these unresolved issues, we designed a paradigm to elicit switching between externally and internally oriented attention. We specifically selected two tasks that were previously successfully applied to induce DMN/DAN (de)activations, namely a visual search task[25] and a self-episodic memory retrieval task[27]. We combined the tasks in a serial order, i.e. one task would follow the other immediately after its completion, hence requiring

immediate switching of attention from the external to the internal mode (and vice versa).

To capture the fast temporal dynamics of the underlying neuronal processes, we measured iEEG during attention switching in a relatively large cohort of 25 subjects with hundreds of electrode contacts localized to the DMN and DAN. The iEEG has the profound advantage of sampling local brain activity with a high temporal (millisecond) resolution and precise (millimeter) anatomical localization of the electrode contacts directly in the human brain. However, iEEG also has costs, including spatially sparse and inhomogeneous sampling across subjects—typically epilepsy surgery candidates[38]. Another advantage of iEEG signals (as compared to the more common scalp EEG, for example) is a reliable signal-to-noise ratio with high frequency power (>50 Hz), such as in the high-gamma band (HGB), which is often regarded as a proxy for the firing rate of the local neuronal population[39,40]. Here, we analyzed the entire frequency spectrum of the iEEG signals from the DMN and DAN during the switch between tasks requiring externally and internally oriented attention. The DMN and DAN were defined based on the available parcellation of the human brain from resting-state functional connectivity from fMRI signals, the so-called Yeo-7 neural networks[41,42], similar to the methodology of recent iEEG studies[33,43].

Below, we detail the electrophysiological description underlying the network transitions between the activated and deactivated states of the DMN and DAN. We analyzed their spectral power changes related to attention switching, the timing of the network activity reversal in different frequency bands, and their interactions by means of functional connectivity.

## Results
### Behavioral results
All 25 subjects with implanted iEEG electrodes (see Suppl. Table 1 and Suppl. Fig. S1 for more details) were able to perform the attention-switching paradigm consisting of a sequence of two tasks: external/internal attention task (Fig. 1a, b; see Methods for more details). From the pseudorandom sequence of both tasks, we extracted the switch trials of the following conditions: internal-to-external and external-to-internal attention switching (Fig. 1c). The error rate for the external attention task was very low (3 ± 1%, mean ± SEM over 25 subjects). The error rate was measured only in the external attention task (the visual search task), for which the correct answer was available to the experimenters (unlike for the internal attention task, a self-episodic memory retrieval task, for which only the subjects knew the correct answer). The mean reaction times (RTs, mean ± SEM) in the switch trials were: internal-to-external RT = 2.8 ± 0.2 s and external-to-internal RT = 2.4 ± 0.3 s. In the analyses below, we focused on the time period around task switching over the interval $t \in [-2.3, 2.3]$ s, where $t = 0$ s indicates completion of the previous task and the simultaneous start of the new task.

**Fig. 1 | External and internal attention switching paradigm.** The paradigm consisted of two successive tasks: **a** A visual search task requiring externally oriented attention. The subjects were asked to find the letter T among many Ls and respond with a gamepad button press (BP) to indicate whether the T was in the upper or lower half of the square. **b** A self-episodic memory retrieval task requiring internally oriented attention, in which the subjects were asked to answer (using the gamepad) yes/no based on whether they agreed (or not) with a statement regarding events from their recent past. From the task sequence **c**, we extracted the switch trials (black frames): either external-to-internal (attention oriented first externally and then internally) or internal-to-external (vice versa). Note that the next task began immediately after the completion of the previous one, i.e., right after the button press (BP, trial time $t = 0$ s).

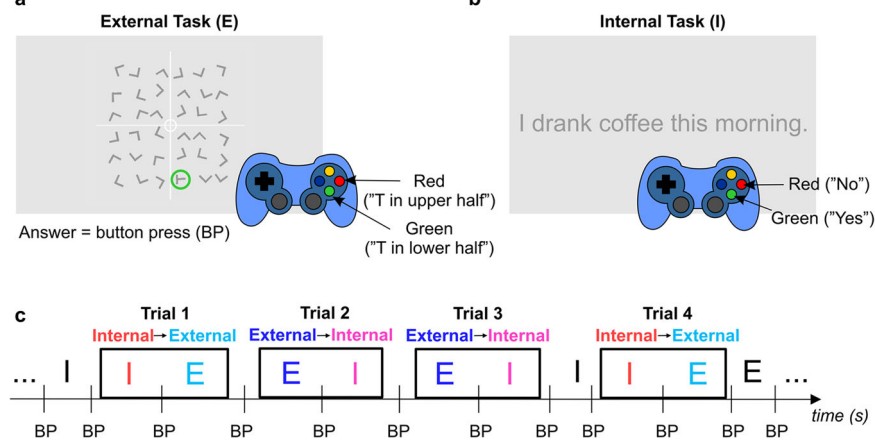

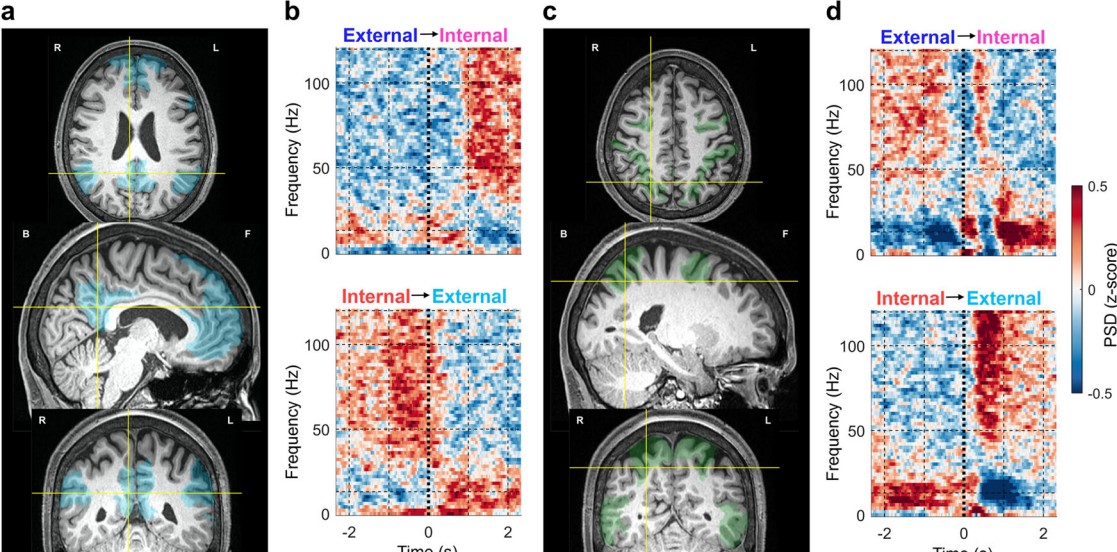

**Fig. 2 | Spectrograms of channels localized to the DMN and DAN during attention switching.** These two exemplary iEEG channels were simultaneously recorded from the same subject (P8). **a** Selected iEEG channel located in the DMN (specifically in the posterior cingulate cortex). The iEEG channel location (yellow crosshairs) is shown in subject-specific brain slices (axial–top, sagittal–middle,

coronal–bottom). **b** Trial-averaged spectrograms ($N_{trials} = 60$) for external–internal attention switching (upper plot) and internal–external attention switching (lower plot). The task switching occurred at time $t = 0$ s. **c, d** Same notation as in a and b for a selected channel from the DAN (specifically in the intraparietal sulcus). The DMN and DAN channels show nearly opposite activation patterns.

## Single-channel examples of iEEG power modulation

First, we illustrate the dynamics of iEEG signal power for two selected channels from DMN (Fig. 2a, b) and DAN (Fig. 2c, d) core nodes in a single subject (P8). Note that we refer to a bipolar-referenced pair of neighboring electrode contacts simply as a "channel" (see Methods for more details). The spectrograms of trial-averaged iEEG brain activity during attention switching enabled three interesting observations: (1) spectral changes in the time domain, i.e. a change in iEEG activity following task switching ($t = 0$ s); (2) spectral changes in the frequency domain between low-frequency power (<30 Hz) and a broadband high-frequency power (>50 Hz); and (3) a nearly opposite pattern of activation between the DMN and DAN. The changes could be observed even in single-trial recordings of raw iEEG (for two exemplary trials measured by the two channels, see Suppl. Fig. S2).

## Spectral power changes of the DMN and DAN

Here, we investigated the spectral power changes of the DMN and DAN at the network level (Fig. 3). We computed the network spectrograms using short-time Fourier transformation (STFT; see Methods for more details) as a grand average across all trial-averaged spectrograms from channels assigned either to the DMN (number of channels $N_C = 741$, number of subjects $N_P = 24$) or to the DAN ($N_C = 297$, $N_P = 24$). Both networks showed the antagonistic activations described above, but demonstrated here at the network level. Note that, at this stage, we made no channel selection other than that based on their localizations (Fig. 3a, c). PSD for the external and internal attention tasks of both networks can be found in Suppl. Fig. S3.

In the subsequent analyses, to quantify the relative band power (RBP) changes between the attention switching conditions, we divided the time-resolved power spectrum into six non-overlapping frequency bands: delta (0.1–3 Hz), theta (4–7 Hz), alpha (8–12 Hz), beta (13–30 Hz), the low-gamma band (31–48 Hz) and the HGB (52–120 Hz).

## The DMN during attention switching

A consistent and interpretable pattern of activations was observed in the DMN (Fig. 4). The large-scale DMN comprised 741 channels (from 24 subjects). As the spatial distribution of the "significant channels" (i.e., channels, where the RBP between the attention-switching conditions was significantly different; see Methods for more details) could be, in principle,

quite different for each of the six aforementioned frequency bands, we also provided their spatial topologies (Fig. 4a, b).

The most pronounced differences between the attention switching conditions (Fig. 4c) were found in the delta, theta, alpha and beta bands. Note that we refer to these bands as the low-frequency bands (<30 Hz). The ratio of significant channels was more than 50% in the delta, theta and alpha bands. We observed significant differences between the internal and external attention tasks ($P < 0.001$, false discovery rate (FDR) corrected for multiple testing across the trial time and the different frequency bands). For the internal attention task, we observed an increase in the HGB and a decrease in the low-frequency bands (and vice-versa during the external attention task).

The relative increase in the HGB, and the concurrent decrease in the low-frequency bands, may be interpreted as a sign of cortical activation. In contrast, the relative decrease in the HGB, and concurrent increase in the low-frequency bands, may be interpreted as a sign of cortical deactivation/inhibition/idling (see Discussion for more details).

An interesting activation pattern, especially in the context of the opposite spectral power changes in the low-frequency bands and HGB, was found in the low-gamma band. While in the other bands (the low-frequency bands and HGB) there was a reversal in the activity (around $t = 0.6$ s) of the external-to-internal and internal-to-external conditions after task switching, the low-gamma band activity strongly and significantly decreased, but only for the internal attention task after the task switch (not for the internal attention task before the trial switch). The low-gamma band activity was thus highly asymmetric with respect to the crossover point, unlike the activity of the other bands. The ratio of significant channels in the low-gamma band was also the lowest of all the bands (32%). The functional significance and interpretation of this pattern of activity in the low-gamma band are less clear.

We also subtracted the non-specific response (common to both conditions; Fig. 4c, black dashed curve) to better visualize the differences and timing of both attention switching conditions (Fig. 4d). Thus the resulting curves should not be directly interpreted in terms of activation/deactivation. Rather, they provide a better visualization of the antagonistic activity and the timing of activity reversal during attention switching in the absence of the common, non-specific activity (which was related to the concurrent visual presentation of a new stimulus and a button press at time $t = 0$ s).

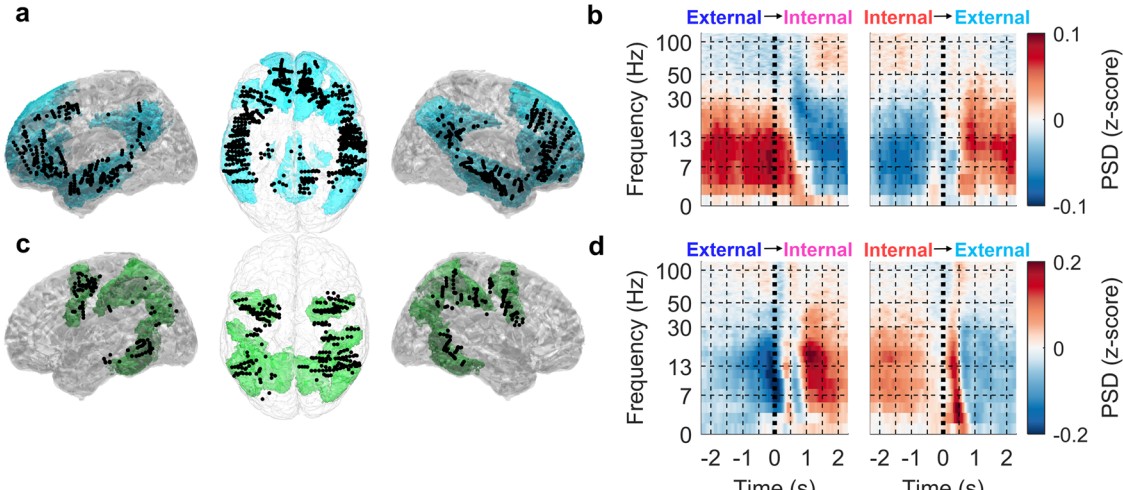

**Fig. 3 | Spectral changes of the DMN and DAN during attention switching.** The DMN and DAN were defined based on the Yeo-7 brain parcellation. **a** Left lateral, top and right lateral views. The iEEG channels (black dots) localized to the DMN (highlighted in blue) drawn on the MNI brain template (colin27, gray-scale). Note that the channels' positions were projected up front in each view for a better visualization (in reality, they were entirely buried deep inside the brain). Left/right lateral views contain channels from only the left/right hemispheres. **b** Grand average spectrograms of all iEEG channels assigned to the DMN (741 channels, 24 subjects).

Each spectrogram represents time (*x*-axis) and frequency (*y*-axis) power spectral density (PSD; color-coded). Note the non-linear, log-like scale of the frequency axis. Left: external-to-internal attention switching (vertical dotted line at *t* = 0 s indicates the time of the task switching). Right: internal-to-external attention switching. **c, d** same notation as in a and b but for the DAN (highlighted in green; 297 iEEG channels, 24 subjects). For both networks, the most pronounced differences between external and internal attention switching can be observed for low-frequency power (<30 Hz).

In the above analysis, we treated the DMN as a unified, monolithic network, which may be an oversimplification[16]. To gain further insight into the behavior of this network during attention switching, we exploited the Yeo-17 parcellation of the human brain, in which the brain is divided into 17 networks and the DMN itself into four subnetworks: DMN A–D (see Suppl. Figs. S4–S7). Despite subtle differences, all DMN subnetworks exhibited behavior consistent with the overall DMN activations.

### The DAN during attention switching
For the DAN, an opposite pattern of activations than for the DMN was observed (Fig. 5). The DAN was sampled by 297 channels from 24 subjects. The largest difference between the attention switching conditions at the group level was again found in the low-frequency bands. The low-frequency band power in the DAN during the external attention task was attenuated and, conversely, increased during the internal attention task. The HGB demonstrated an opposite pattern of activation, with an increase in activity during the external attention task and a relative decrease during the internal attention task, although the differences between the attention-switching conditions were non-significant (*P* > 0.001).

There was a strong transient activity in the HGB after the presentation of the new stimulus presentation (*t* = 100–200 ms). We hypothesize that these rapid changes in HGB activity correspond to the button press at *t* = 0 s, when the DAN was briefly deactivated and the motor cortex was activated. Concurrent with the button press was the presentation of the new stimulus, resulting in a rapid increase in the HGB with a peak around *t* = 500 ms.

Curiously, although the majority (70%) of the channels in HGB displayed significant difference between the attention switching conditions (i.e., significance on a trial level), there were no significant differences at the network activity level, suggesting heterogeneous responses in this frequency band among the channels. The low-gamma band (31–48 Hz) had a similar temporal profile to the power in the low-frequency bands. Utilizing the channel assignment into the Yeo-17 atlas[41], we split the DAN network into two subnetworks: DAN-A and DAN-B (Suppl. Figs. S8 and S9). Although the low-frequency band activations were quite consistent with the overall pattern in the DAN, the HGB exhibited differential responses in the two subnetworks without significant differences in the attention-switching conditions of each subnetwork.

Due to the transient, evoked activity connected to the stimulus presentation, it was not always apparent when the switching occurred. To better illustrate this (Fig. 5d), we subtracted the common activation (i.e., common to both attention-switching conditions)—the dashed black curve in Fig. 5c.

### Strength of the RBP difference during attention switching
In summary, so far, we have confirmed the antagonistic behavior of the two networks in terms of their activations/deactivations during attention switching. A novel observation was the robust and highly reproducible neural dynamics of the attention switching in the low-frequency bands (<30 Hz), which have received less attention in the literature than the HGB. To verify the visual observation that the largest differences in attention switching were in the low-frequency bands, we computed the mean "strength of the RBP difference" between the conditions (see Methods), as a mean across time for each channel and frequency band (Fig. 6a). For the DMN, the strength of the RBP difference between both conditions was highest in the delta, theta and alpha bands. The differences in these low-frequency bands were significantly stronger than those in the HGB (*P* < 0.001, two-tailed Wilcoxon rank sum test). For the DAN, the most pronounced differences were also found in the low-frequency bands (alpha, delta and theta), albeit not significantly larger than in the HGB (*P* > 0.001).

Note that in the above analyses, we did not perform any trial or epoch rejection. To mitigate possible concerns about the role of artifacts in the iEEG data, we repeated the RBP analysis including the data rejection (see Methods for more details). The RBP activations of the DMN (Suppl. Fig. S10b) and DAN (Suppl. Fig. S11b) were almost identical, which can be explained by the large number of trials and channels comprising the network-level activity and the rather low susceptibility of iEEG to artifacts in general[44]. On a similar note, to ensure that the RBP was correctly (albeit not necessarily optimally) estimated by the STFT with variable window size, we repeated the RBP analysis using the Morlet wavelet transformation (see Methods for more details), leading again to very similar results (correlation coefficient between the RBP estimated by STFT and the wavelet transform was 0.90 ± 0.05 for DMN and 0.93 ± 0.03 for DAN; mean ± SEM across all frequency bands and conditions; Suppl. Fig. S10c and S11c).

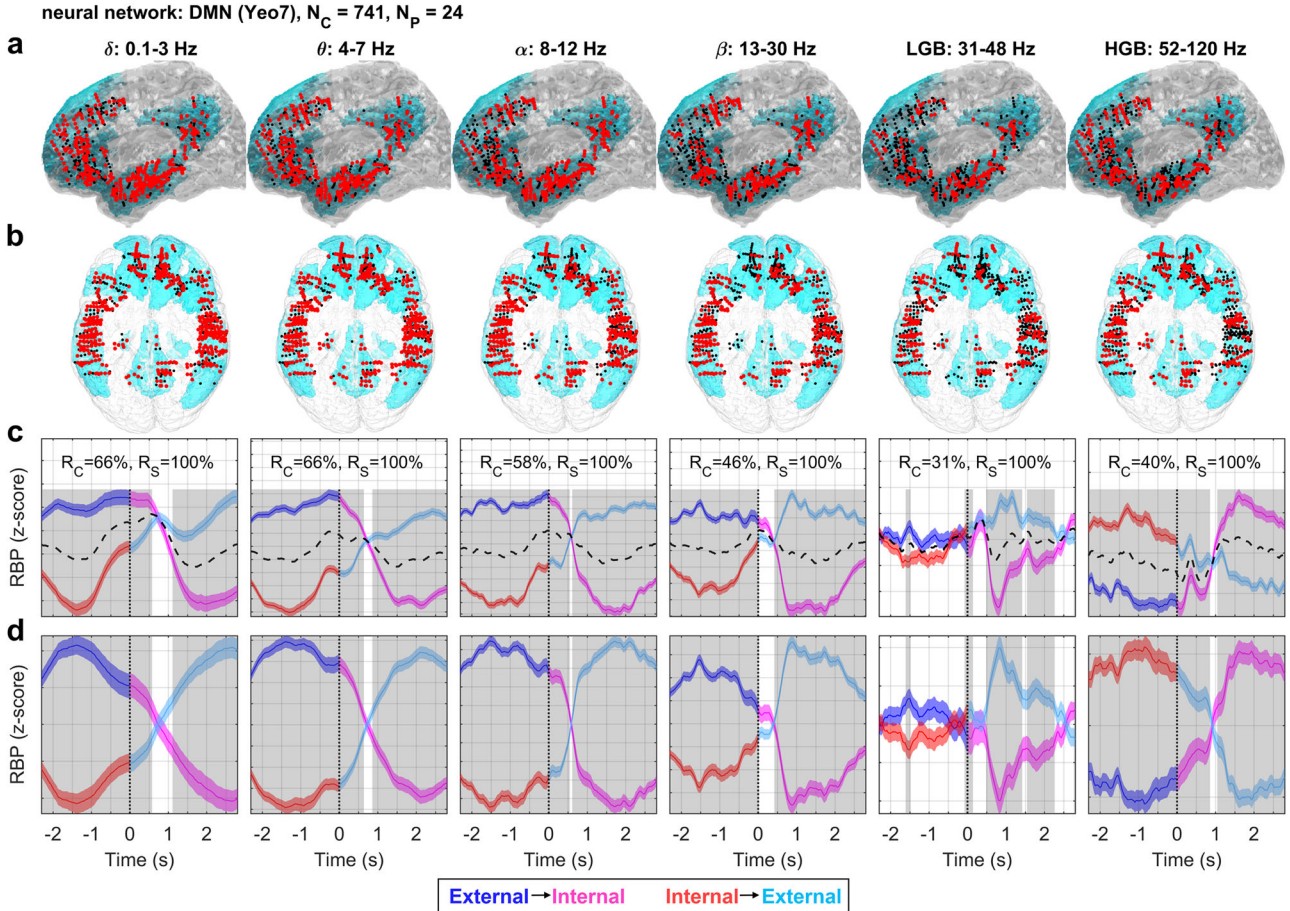

**Fig. 4 | DMN activity during attention switching.** The total number of iEEG channels assigned to this network was $N_C = 741$ from $N_P = 24$ subjects. Columns: RBP in different frequency bands indicated by the title. Rows: **a** Left lateral and **b** top views of iEEG channels projected onto the MNI brain template with the DMN network highlighted in light blue. Note that for the left lateral view, channels from both hemispheres were projected up front for a better visualization. Each iEEG channel was classified either as significant (i.e., having significantly different activity between the trials of both attention switching conditions, $P < 0.05$, FDR corrected) (larger red dots) or non-significant (smaller black dots). **c** Each subplot: Grand average of RBP modulation ($y$-axis) in time ($x$-axis) for the external-to-internal (blue–magenta curve) or internal-to-external (red–cyan curve) attention switching conditions computed across all significant channels (plotband: mean ± SEM). The logic behind this color change was to accentuate the task switch; the reddish colors (magenta and red) indicate the internal attention task while the blueish colors (blue and cyan) the external attention task. Time $t = 0$ s (vertical dotted line) represents the time of task switching. The non-specific activity (dotted black curve) was computed as a mean across both conditions. Significance of the difference between the attention-switching conditions at the network level (highlighted in gray) was assessed over the trial-averaged channels comprising the network activations ($P < 0.001$, FDR corrected). The $y$-scale was adjusted for each subplot (note the $y$-grid spacing, where the offset of each grid line equals 0.025 of the $z$-score). $R_C$ = ratio of significant iEEG channels in each neural network; $R_S$ = ratio of different subjects with at least one significant channel. **d** Each subplot: RBP modulation relative to the non-specific activations. The non-specific activations, common to both conditions (dashed black curves in C), were subtracted from both the external-to-internal (blue–magenta curve) and internal-to-external (red–cyan curve) conditions, highlighting the timing of the switching pattern in each frequency band. There was a robust reversal pattern for the low-frequency power (<30 Hz) with inverse HGB activations.

## Timing of the crossover point of neuronal activity

For each network and each RBP, we investigated if there was any difference in the timing of the RBP crossover during attention switching (Fig. 6b). Our hypothesis was that a faster recruitment of the network leads to a faster crossover (or reversal) of its activity (see Methods and Suppl. Fig. S12 for more details). There were significant differences in the distribution of the crossover points in the HGB, when comparing DMN and DAN ($P < 0.001$, two-tailed Wilcoxon rank sum test, FDR corrected for multiple testing across the different frequency bands), but—surprisingly—not in the low-frequency bands. The mean difference of the crossover points between the DMN and DAN in the HGB was 220 ms, well in line with the results of other studies on timing differences between the DMN and DAN (see Discussion for more details).

## Functional connectivity between the DMN and DAN

Finally, we investigated the functional connectivity between the DMN and DAN (Fig. 7). We selected two different, data-driven and commonly used

approaches: a non-directed connectivity measure, the phase-locking value (PLV), and a directed connectivity based on multivariate autoregressive (MVAR) model: the directed transfer function (DTF), the partial directed coherence (PDC) and the Granger–Geweke causality (GGC). Given the large number of connectivity estimators, one of the reasons for selecting this particular set lied in their different characteristics: the PLV is solely based on phase synchronization of iEEG signals, while the directed connectivity measures based on the MVAR model take into account both phase and amplitude of the signals.

The functional connectivity was assessed in the different frequency bands between all pairs of channels from the DMN and DAN for each subject individually (see Methods). We selected only those subjects with at least five significant channels in both networks (to exclude subjects with very sparse network sampling). The channels could be significant in any of the six frequency bands. These criteria were met in 17 subjects, including 465 channels in the DMN ($N$ channels/subject = 27 ± 3, mean ± SEM) and 235 channels in the DAN ($N$ = 14 ± 2) (Fig. 7a). The

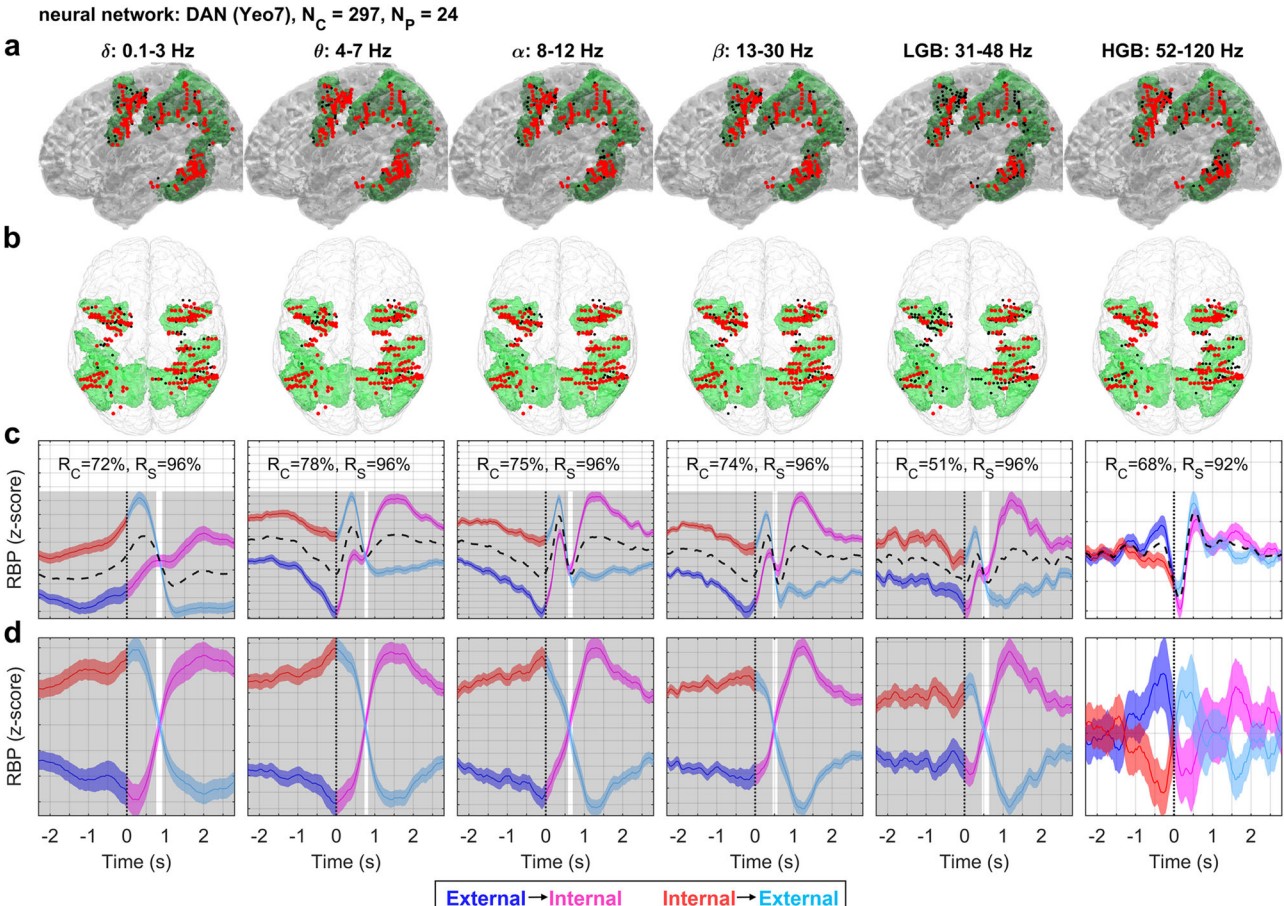

**Fig. 5 | DAN activity during attention switching.** The total number of iEEG channels (neighboring bipolar contact pairs) assigned to this network was $N_C = 297$ from $N_P = 24$ subjects. Same conventions as in Fig. 4. Columns: RBP in different frequency bands indicated by the title. Rows: **a** Left lateral and **b** top views of iEEG channels projected onto the MNI brain template with the DAN network highlighted in green. Each iEEG channel was classified either as significant (i.e., having significantly different activity between the trials of both attention switching conditions, $P < 0.05$, FDR corrected) (larger red dots) or non-significant (smaller black dots). **c** Each subplot: Grand average of RBP modulation (y-axis) in time (x-axis) for the external-to-internal (blue–magenta curve) or internal-to-external (red–cyan curve) attention switching conditions computed across all significant channels (plotband: mean ± SEM). Time $t = 0$ s (vertical dotted line) represents the time of task switching. The non-specific activity (dotted black curve) was computed as a mean across both conditions. Significance of the difference between the attention-switching conditions at the network level (highlighted in gray) was assessed over the trial-averaged channels comprising the network activations ($P < 0.001$, FDR corrected). The y-scale was adjusted for each subplot (note the y-grid spacing, where the offset of each grid line equals 0.025 of the z-score). $R_C$ = ratio of significant iEEG channels in each neural network; $R_S$ = ratio of different subjects with at least one significant channel. **d** Each subplot: RBP modulation after subtraction of the non-specific activations, common to both conditions (dashed black curves in (**c**)). There was a robust reversal pattern, especially for low-frequency power (<30 Hz), inverse to the DMN activity.

significance of the difference between attention-switching conditions was assessed across the subjects (comparing 17 mean DMN values of each subject between the two conditions) at each time step ($P < 0.001$, FDR corrected for multiple testing across time steps and frequency bands).

Significant PLV differences between attention switching conditions were found only in the delta frequency band (Fig. 7b), where the PLV for the internal attention task was significantly higher than for the external attention task. Another pronounced PLV modulation was found in the theta band, with a clear peak at $t = 0.5$ s, matching the crossover points of the DTF (Fig. 7c, d) and slightly preceding in time the crossover point of the RBP in this frequency band (comparing the theta band in Fig. 6b and Suppl. Fig. S13a). The higher frequency bands showed desynchronization with much smaller PLVs. Higher PLVs in the low-frequencies (0–7 Hz) may be suggestive of inter-network communication at the time of attention switching.

The DTF (and PDC as well as GGC) was computed in sliding windows to capture the temporal dynamics by fitting a multivariate autoregressive (MVAR) model at the single-subject level. We first confirmed that the

MVAR model was able to fit the RBP changes (see Suppl. Fig. S13): the correlation coefficient between the RBP estimated by STFT and the MVAR model was $0.95 ± 0.01$ (mean ± SEM across the six frequency bands and both networks). The task-related DTF changes were broadband and largely in low-frequency bands (Fig. 7c, d), for which the DTF from the DMN to the DAN ($DTF_{DMN \rightarrow DAN}$) was significantly increased during the internal attention task and decreased during the external attention task (Fig. 7c), with the crossover around $t = 0.5$ s. The opposite pattern was observed for the $DTF_{DAN \rightarrow DMN}$, for which higher values were found during the external than during the internal attention task (Fig. 7d). There was little to no DTF modulation in the HGB. Similar results were observed for the PDC (Suppl. Fig. S14) and GGC (Suppl. Fig. S15).

Interestingly, in the alpha and theta bands, the timing of the DTF crossover between attention-switching conditions preceded by approximately 100 ms the timing of the RBP crossover estimated by the same MVAR model (comparing Fig. 7c, d and Suppl. Fig. S13b, d). The results suggest a stronger direction of information flow from the activated network to the deactivated network than the other way around (see Discussion for more details).

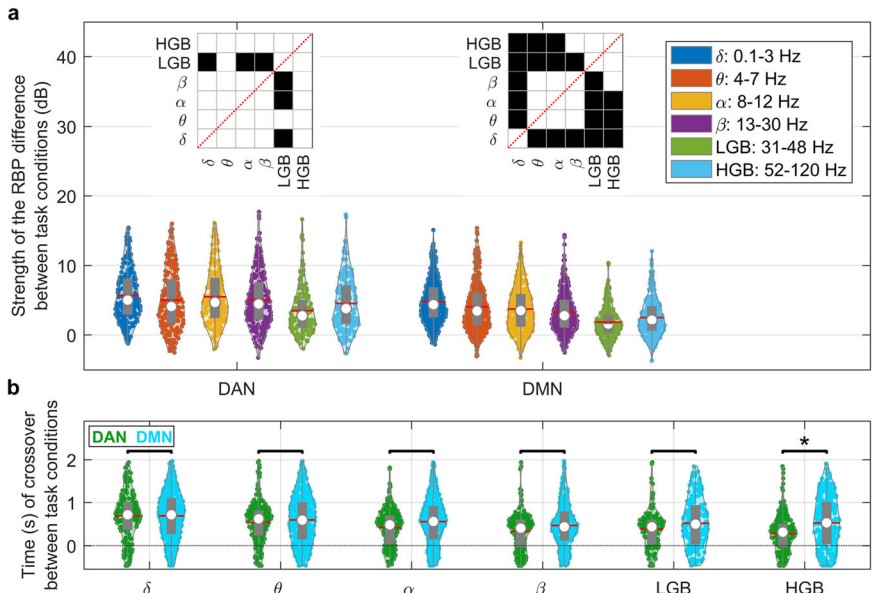

**Fig. 6 | Strength of the RBP difference and timing of crossover between externally and internally oriented attention. a** The strength of neural representation during attention switching. The strength of the RBP difference (in dB) between external-to-internal and internal-to-external attention switching was computed across all significant channels in a given frequency band as a log-transformed mean across the entire time interval from −2.3 s to 2.3 s. The width of the violin-like plots represent the data distribution across channels (colored dots), mean is denoted by red line, median by white circle and gray boxes represent the interquartile range. In both networks, the largest strength of difference was observed in the power of the low-frequency bands (<30 Hz), but also in the HGB (52–120 Hz). A small strength of the RBP difference was observed in the low-gamma band (LGB, 31–48 Hz). The insets above error bars represent significance of difference (black: $P < 0.001$; white: $P > 0.001$, FDR corrected) between the different bands assessed by a Wilcoxon rank sum test of the trial-averaged channel RBP activities. **b** The distribution of crossover points of neural activity between external-to-internal and internal-to-external attention switching for the DAN (green) and the DMN (cyan). Same notation for the violin-like plots as in a, each dot represents a single-channel cross-over time of RBP between the task conditions. The switching occurred significantly faster ($P < 0.001$, FDR corrected) in the DAN than in the DMN, but only in the HGB.

## Summary of the main results

We summarize the main results in a simple diagram (Fig. 8). When the network was deactivated, its activity was dominated by slower frequencies, while in its activated state, the iEEG oscillations were much faster (these changes could be observed even in single-trials, see Suppl. Fig. S2). In a schematic diagram, we illustrate the direction of the information flow as determined by the directed (DTF) connectivity (Fig. 8, red arrows) as well as the non-directed, functional connectivity measured by the PLV (Fig. 8, blue double arrows). The direction of the information flow (in frequencies <50 Hz) was from the activated network to the deactivated network. At the time of the network activation reversal, we observed higher PLVs for the lowest frequencies (0–7 Hz), suggesting possible inter-network communication.

## Discussion

This study describes the transition between activated states of the DMN and DAN during switching between external and internal attention, as measured by iEEG signals. Most robust representation of attention switching was found in the low-frequency bands (<30 Hz), especially in the alpha (8–12 Hz) and theta (4–7 Hz) bands (Fig. 6a). Low-frequency band power is often overlooked in iEEG studies, many of which focus exclusively on the HGB[25,27,34,35]. The decrease in the low-frequency bands was typically accompanied by a simultaneous increase in the HGB power and was most pronounced in the DMN (Fig. 4). Such a simultaneous change in the spectral power between the low and high frequencies is often interpreted as a general marker of cortical activation[45,46]. Interestingly, in our context, the simultaneous decrease in the low-frequency bands and increase in the HGB has been suggested to be associated with an increase in attention or task engagement on a behavioral level[47].

A possible functional role of the alpha oscillations is that the attenuation of their rhythmic amplitude increases network excitability[48], and thereby promotes its representational and computational capacity,

which could, in turn, be manifested by the simultaneous increase in the HGB[47]. Broad-band HGB activity is commonly interpreted as a proxy for the aperiodic firing (or multi-unit activity) of the underlying neuronal population[39,40]. Put the other way around, an increase in the alpha power synchronizes the network's activity, rendering it inactive/idling[49]. It has also been suggested that alpha oscillations serve as a gating mechanism for inhibitory control, in which the increase in the alpha power is associated with the gating of task-unrelated areas, whereas a decrease reflects their release from the inhibition[48,50]. Interestingly in this context, also the fMRI BOLD signal was found to be anticorrelated with EEG alpha power[22,51].

Given this context, our interpretation of the network activations is the following: a relative increase in the HGB and a simultaneous decrease in the low-frequency bands implicates network activation and information processing, while a decrease in the HGB and an increase in the low-frequency bands renders the network deactivated (inhibited, or idling). An interesting activation pattern was observed in the low-gamma band (30–48 Hz) of the DMN, where there was a sharp decrease in the low-gamma band power exactly at the time of attention switching ($t = 0.5–1.0$ s), but only for the external-to-internal attention switching condition (Fig. 4c). The interpretation of low-gamma band activity remains unclear. The low-gamma band could play a functional role in mediating the network activations (as a bridge between the low-frequency bands and HGB) and interactions (e.g., the binding-by-synchrony hypothesis)[52].

The presented results are well in line with previous studies on DMN (de)activations using fMRI[3,5,9] and iEEG[23–30,32–36]. Here, we also observed a significant decrease in the HGB of the DMN during the external (as compared to the internal) attention task (Fig. 4), and the opposite during the internal attention task. The internal attention task activations were most pronounced in the "DMN-B subnetwork" (based on the Yeo-17 brain parcellation, see Suppl. Fig. S5), comprising the middle and superior temporal gyrus. These DMN areas have been implicated in language comprehension and semantic processing[53], here reflecting the nature of the stimulus

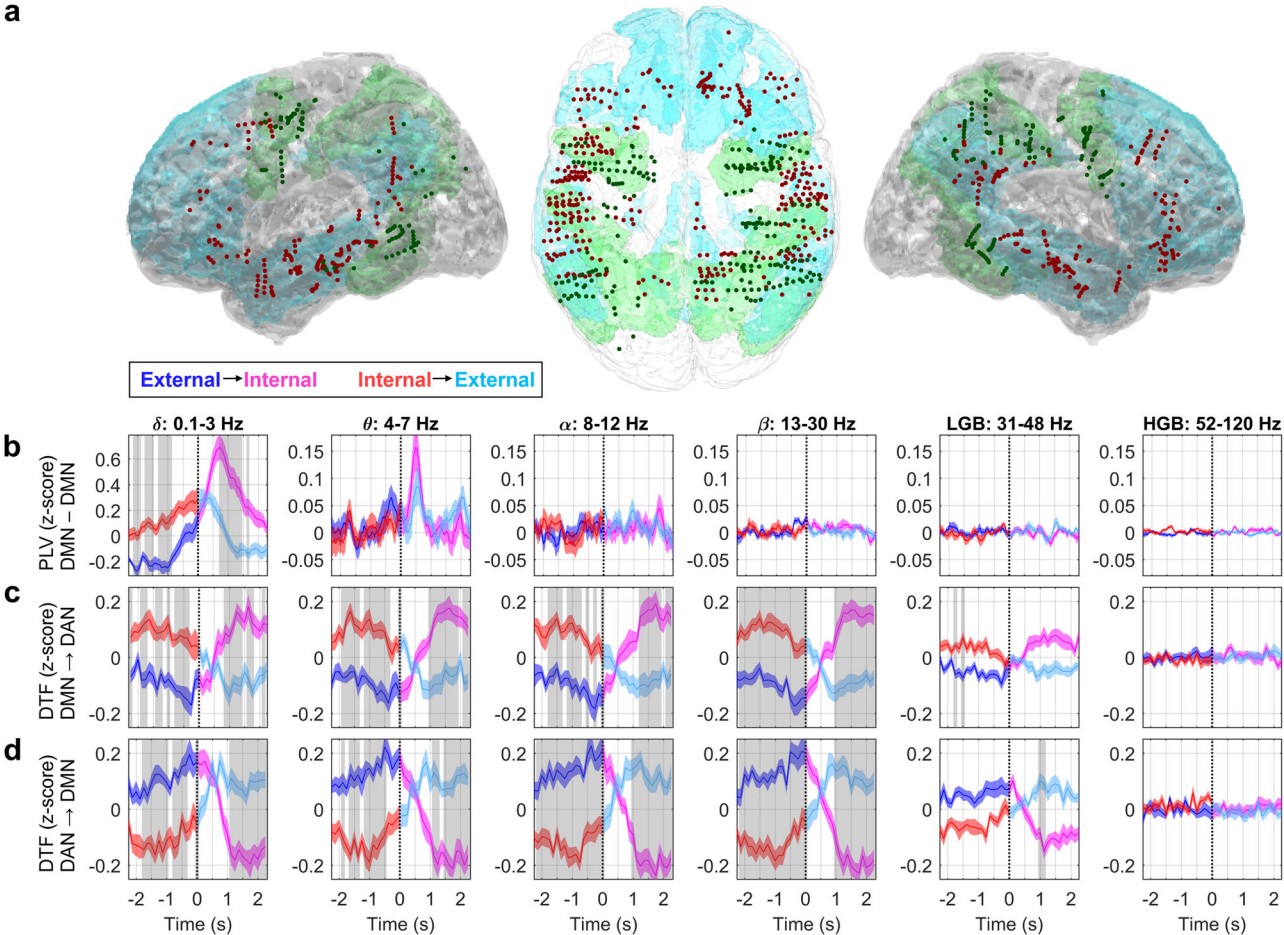

**Fig. 7 | Functional connectivity between the DMN and DAN during attention switching.** The functional connectivity was evaluated by the PLV and by the DTF. Seventeen subjects with at least five significant channels in each network were included in the analysis. **a** Left lateral, top, and right lateral projections of iEEG channels onto the MNI brain template with highlighted neural networks (DMN–blue, DAN–green) and significant channels (DMN—dark red, DAN—dark green dots). **b** The PLV between the DMN and DAN in the six different frequency bands (indicated by their titles). Significant differences in distributions between the attention-switching conditions ($P < 0.001$, FDR corrected) marked by gray rectangles. Each subplot: Grand average of temporally resolved (x-axis) PLV (y-axis) for the external-to-internal (blue–magenta curve) or internal-to-external (red–cyan curve) attention-switching conditions (plotband: mean ± SEM). Time $t = 0$ s (vertical dotted line) represents the time of task switching. **c, d** The directed connectivity was measured by the DTF from the DMN to the DAN (**c**) and from the DAN to the DMN (**d**) for both attention-switching conditions. Same notations as in (**b**). Both the PLV and DTF showed the most robust effects in the low-frequencies (<30 Hz), suggesting their functional relevance in inter-network communication.

presentation of the internal attention task (including not only the self-episodic memory retrieval but also the semantic comprehension of a written sentence).

Using the fast temporal resolution of the iEEG, several studies have investigated the temporal order of networks activations and deactivations. For example, Raccah et al.[34] showed that activations in the superior parietal lobule (a core node of the DAN) preceded by approximately 200 ms the deactivations in the posteromedial cortex (core region in DMN) during arithmetic tasks. Similarly, Kucyi et al.[33] found that the responses peaked first in the DAN and then approximately 300 ms later in the DMN during a gradual onset continuous performance task. In a recent study on mentalizing about self and others (i.e., theory of mind), Tan et al. demonstrated that activations began in the visual cortex, followed by DMN regions in the temporoparietal cortex (lagging by approximately 200 ms) and, even later (300–400 ms), the medial prefrontal regions[35]. While these studies investigated activation onsets (or peaks), our methodology was different, as we focused on the crossover points of neuronal activity during attention switching.

We found that the crossover point occurred about 200 ms earlier in the DAN than in the DMN—a value well in line with the above reports. Interestingly, the timing difference between DMN and DAN was observed only in the HGB (and also in the delta power), but not in the theta, alpha, or beta bands (Fig. 6b), suggesting that they may play different functional roles in the networks' (de)activations. Taken together, these data support the idea that the transmodal DMN integrates the information on a slower timescale than the other networks[17] and that it is positioned at the top of the information-processing hierarchy of brain networks[54].

Apart from descriptions of the DMN activations, some—albeit considerably fewer—iEEG studies have also investigated interactions between the DMN and other networks. The network interactions are typically assessed by means of either directed (i.e., including direction/causality of interactions) or non-directed functional connectivity (see[55,56] for recent reviews). For example, Foster et al.[29] found that the intrinsic iEEG connectivity patterns observed in slow fluctuations (<1 Hz) of HGB activity of selected DMN regions were highly correlated with those obtained from resting-state fMRI from the same subjects. In a recent study, Das et al.[43] investigated both functional connectivity by PLV and the phase transfer entropy both within DMN and with all other Yeo-7 networks. They found higher values of phase transfer entropy for the DMN's inter-network connectivity than for intra-DMN connectivity during a free recall memory task, suggesting that the DMN is a causal outflow network.

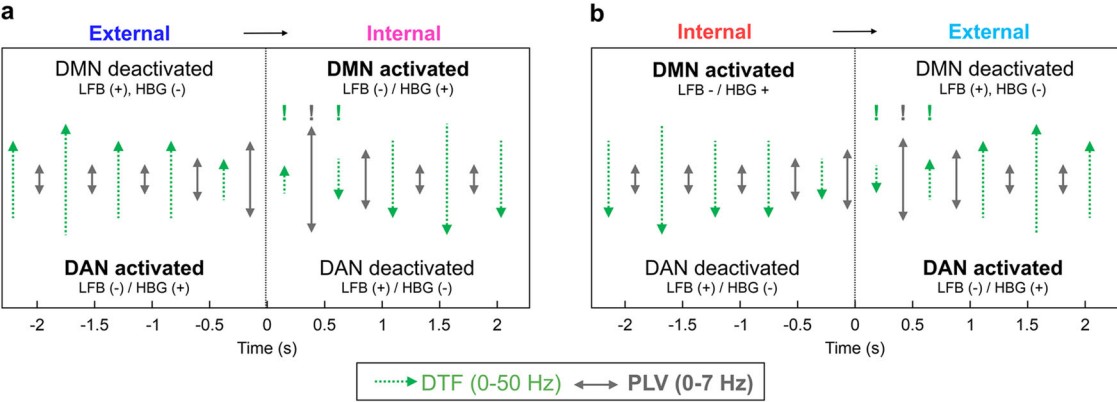

**Fig. 8 | Schema of DMN and DAN activations and interactions during attention switching. a** A summary diagram of the main results for external-internal attention switching. Network activation/deactivation is indicated in the text of the diagram. When a network was activated (in bold font), we observed a power increase (+) in the HGB and a decrease (−) in the low-frequency bands (LFB) (and vice versa for a deactivated network). Network connectivity is schematically illustrated by the arrows (their lengths approximate the difference between the attention-switching conditions: the longer the arrow, the larger the connectivity value). The dotted green arrows indicate the directed connectivity results from the DTF in the frequency range 0–50 Hz (reversal of the information flow highlighted by exclamation marks above). The gray double arrows indicate the functional (non-directional) connectivity measured by the PLV in the range 0–7 Hz (the peak highlighted by an exclamation mark). **b** A summary diagram of the main results for internal–external attention switching. Same notations as in a. The changes in network activations were accompanied by changes in the direction of the information flow between them (DTF) and higher functional connectivity values (PLV) at the time of the reversal could then indicate higher inter-network communication.

In our study, we used the PLV and the DTF to estimate the functional connectivity during attention switching. First, we would like to emphasize that the interpretation of connectivity measures is still controversial[57–59]. Here, we adhere to the following notions: an increase in the PLV is suggestive of inter-network communication, as two communicating network nodes presumably result in transient, frequency-specific phase synchronization[60]. For the DTF, if $DTF_{m \to n}$ is greater than $DTF_{n \to m}$, then the dominant direction of information flow is from network node $m$ to network node $n$[61]; hence, node $m$ exerts a larger causal influence on node $n$.

The PLV between the DMN and DAN exhibited a plausible temporal profile only in the lowest frequency bands (i.e., the delta and theta bands; Fig. 7b), supporting their functional role in inter-network communication. Interestingly, it has been suggested that theta and delta oscillations facilitate phase coding in the human hippocampus[62], and, recently, that the connectivity within the DMN itself is dominated by slow-wave synchronization[43].

Our data from the time-resolved DTF (Fig. 7c, d), PDC (Suppl. Fig. S14) and GGC (Suppl. Fig. S15) showed a clear reversal in the information flow between the activated and deactivated states of the DMN and DAN during attention switching. When the network was activated (i.e., the DMN during internal attention task and the DAN during the external attention task), it exerted a higher causal influence on the deactivated network (there was a higher information inflow from the activated to the deactivated network manifested by a significant increase in the DTF). This could be interpreted as the activated network "inhibiting" the deactivated network.

The high similarity among the directed connectivity measures (DTF, PDC and GGC) can be likely attributed to averaging across the different channels of both networks in each subject and also to the fact that they were computed using the same MVAR model. Given that we found little evidence for higher frequency phase synchronization between the DMN and DAN in our PLV analysis (Fig. 7b), the results suggest amplitude-amplitude coupling as the dominant interaction mechanism during the attention switching. A future study could address the phase-amplitude coupling for possible nested oscillations among the different frequency bands, for example, clarifying whether the inter-network communication is established by phase synchronization of oscillations at lower frequencies, acting as a temporal reference frame for information carried by high-frequency activity[63].

Notably, our results do not exclude the possibility that the interactions between the DMN and DAN are mediated by another network. Some studies have suggested that the salience network could act as a switch between the different modes of attention[64], especially in bottom-up attention tasks. Other studies have proposed that thalamocortical circuits could play a major role in regulating the activity of large-scale, distributed networks[65], such as the DMN and DAN.

There are several major limitations of this study. One of the major limitations, inherent to iEEG methodology, is the high spatial variability of electrode placement across subjects. Another weak point is the definition of the spatial extent of the DMN and DAN, here defined by the Yeo-7 atlas[41]. A more accurate delineation of the networks could be achieved by using data from resting state fMRI[29]. As we did not have the resting state fMRI measured for all patients in our cohort, we resorted to the Yeo-7 atlas[41], which can result in loss of some fine-grained details due to spatial blurring when computing population-average maps. Hence, it is quite likely that some (but presumably only a few) channels could have been assigned to the DMN or DAN incorrectly, or could have been missed, by not considering the subject-specific boundaries of the DMN and DAN. However, we are convinced that the number of such misassigned channels is relatively low. Importantly, due to the large amount of data (hundreds of channels from as many as 24 subjects in each network), this imprecision played a marginal role in the reported network activity. Moreover, we treated the DMN and DAN as monolithic networks, which was a useful oversimplification[16]. Future studies could, for example, extend the portfolio of the external and internal attention tasks and investigate the functional stratification of the DMN. Measurement of directed connectivity dynamics by the DTF is limited by a presumption of a closed system. Node-to-node directional connectivity can be influenced by the information entering the system from outside, which increases the total inflow to the node and thus decreases the ratio of node-to-node inflow—seemingly decreasing the DTF. Elimination of this complication is difficult in complex systems such as the brain.

In conclusion, our findings have uncovered the neural activity during attention switching between internal and external modes of perception in two antagonistic brain networks: the DMN and DAN. We highlight the role of low-frequency power modulation, which has often been overlooked, as many previous studies focused on the HGB. We also provide an important insight into the directionality of interactions between the DMN and DAN, showing that the flow of information was from the activated network to the deactivated network, reversing itself in a meaningful way after the attention

switched. We anticipate that further insights into the precise relationship between low-frequency and high-frequency activity, as well as into mechanisms of network inhibition could play a pivotal role, not only in systems neuroscience, but also in the treatment of various psychiatric disorders, in which pathological activity of the DMN has been implicated.

## Methods

### Participants

The subjects ($N = 25$; 15 females; mean ($\pm$SD) age = $34 \pm 12$ years) were patients with drug-resistant epilepsy who were undergoing iEEG video monitoring as a part of presurgical evaluation at Motol University Hospital in Prague, Czech Republic. The number and trajectories of the intracranial electrodes were based solely on clinical needs to delineate the extent of the epileptogenic network. The study was approved by the ethics committee of Motol University Hospital. All ethical regulations relevant to human research participants were followed. The patients participated voluntarily after signing an informed consent form. Details about the subjects and their electrode implantations are summarized in Suppl. Table 1 and Suppl. Fig. S1.

### Attention switching paradigm

The experimental paradigm was designed to investigate neuronal activity during attention switching between external and internal modes of perception. We specifically used switching between two tasks that were previously used to investigate external and internal attention representation: (1) a visual search task requiring externally oriented attention[25] (Fig. 1a) and (2) a self-episodic memory retrieval task requiring internally oriented attention[27] (Fig. 1b).

The goal of the external attention task was to find the letter T among 35 Ls arranged on an 6 × 6 grid with random rotations and to indicate whether the T was in the upper or lower half of the grid by an appropriate button press on a gamepad (Logitech F310). A red/green button press meant that T was in the upper/lower half of the square, respectively (Fig. 1a). In the internal attention task, the subjects were asked to provide a yes/no answer to a given statement based on memories of their own past experiences (i.e., whether they agree with the statement or not). The answer was provided by pressing a button on the gamepad (red = no, green = yes).

As soon as the subjects provided their answers by pressing the red or green button on the gamepad (or in case of a time-out), the next trial started. Hence, there was no pause (e.g. a "hold period" with a fixation cross) between the external and internal attention tasks. Note that the task did not switch on every trial. The subjects were instructed to provide their responses as quickly as possible. The maximum time for their response was set to 5 s. Note that (1) the correct answer was known to the experimenters only in the external (but not in the internal) attention task and (2) the same set of red and green buttons was used for answers in both tasks.

The whole experiment was first explained to the subjects in a short presentation, followed by a few (typically one or two) test sessions, so the subjects felt comfortable with and understood the paradigm. The attention switching paradigm was split into four, several minutes-long recording sessions. There was a pause between the sessions, the length of which depended on each subject individually (typically only a few minutes). The entire paradigm lasted about 30 min. The experiment was implemented in Psychtoolbox-3[66] and synchronized with the iEEG data by sending a trigger mark at the start of each trial into the trigger channel of the recording amplifier via a parallel port.

### iEEG data recording and preprocessing

The iEEG was measured by intracerebral electrodes (DIXI Medical), recorded by medical amplifiers (Quantum, NeuroWorks), sampled at 2048 Hz (bandwidth 0.01–682 Hz) and later downsampled to 512 Hz (to facilitate faster computation). The electrode contacts (cylindrical shape, 0.8 mm diameter, 2 mm height and 1.5 mm inter-electrode contact distance) were collinear and arranged on an electrode shank penetrating the brain parenchyma. The reference (and ground) electrodes were located in white matter (in subject-specific locations). The number of implanted electrodes differed among subjects based on their specific diagnostic requirements.

The downsampled iEEG recordings were first visually inspected and broken channels were rejected. Based on the information from an experienced EEG reader, channels located in the seizure onset zone, irritative zone (i.e., having high inter-ictal epileptic activity), or heterotopic cortex were rejected, as well. The remaining iEEG recordings were analyzed using bipolar referencing, starting from the first (deepest) electrode contact and referenced to the nearest neighbor, similar to our previous study[67]. We refer to a pair of bipolar referenced neighboring electrode contacts simply as the "channel." All iEEG channels were high-pass filtered (0.1 Hz cutoff, Butterworth filter, 6th order, zero phase shift) to remove slow drifts and notch-filtered at 50 Hz and its harmonics to reduce the line noise (48–52 Hz stop band, Butterworth filter, 6th order, zero phase shift). These minimally preprocessed iEEG data, to which we refer as "raw" data, were subject to further analysis, as described below.

### Trial extraction

From the sequence of the tasks (Fig. 1c) and synchronized, raw iEEG data, we extracted non-overlapping "switching trials," or simply "trials," consisting of the following two conditions based on the attention orientation change: (1) from external to internal or (2) from internal to external. We will refer to these as "attention switching conditions" (or just "conditions"). The extracted trials were in the format $D(t,ch,tr_c)$, where $t$ is time, $ch$ channels and $tr_c$ trials of condition $c \in \{external\text{-}to\text{-}internal, internal\text{-}to\text{-}external\}$. The zero time ($t = 0$ s) matched the time of the button press (answer) and also of the stimulus presentation for the new task. Because of further post processing in some analyses and associated edge artifacts, we extracted slightly longer trial periods (from $-4$ s to 4 s) and cropped them for visualization in the Results (from $-2.3$ s to 2.3 s). We extracted 60 switch trials for each condition.

In the presented results, we made no trial rejection. However, a rightful concern could arise to which degree the results could be affected by various artifacts, mostly from the epileptic activity (such as inter-ictal discharges). Hence, we replicated the analyses of RBP activations including data rejection (Suppl. Fig. S10b and S11b). In particular, we rejected epochs of epileptic activity (e.g., inter-ictal discharges) as the epileptic network activity may affect the other, non-seizure related channels as well. To this end, we opted for an epileptic spike detector[68], used in our previous studies[67,69,70]. Additionally, we also rejected those epochs, where the amplitude of the raw iEEG data was above or below six SDs from the mean (defined for each channel and recording session separately). The indices of the detected epochs (of individual duration) with additional, 100-ms long margins were rejected from the analyses, resulting in 0.6 ± 0.1% of rejected data from the trials; mean ± SEM across 25 subjects). Note that we did not reject whole trials, in order to preserve maximum of the data and trials, but only epochs of specific duration.

### iEEG channel assignment

The electrode contacts were localized using post-implantation CT scans coregistered to pre-implantation MRI. The positions of the contacts were also verified based on post-implantation MRI. The MRI scans were normalized to MNI space using SPM12. With this procedure, we obtained the MNI coordinates of the electrode contacts[71]. Note that due to the warping of the brain during MNI normalization, some electrode shanks appeared bent in the figures, while, in reality, they were linear. Each iEEG channel was assigned the MNI coordinate equal to the center of the contact pair. The MNI coordinates were used to localize the channels into the regions of interest (ROIs) defined by the Yeo-7 brain parcellation[41,42]: the DMN or DAN. The Yeo-7 atlas was normalized to the same MNI brain template as the subject-specific MRI scans. We also included channels with a maximum distance of 10 mm between the channel MNI and the nearest voxel of the ROI, which typically concerned channels located in the white matter near the border with the gray matter. We were motivated to allow for this offset by (1) trying to maximize the number of the iEEG channels in the analyses and (2)

results from source localization studies using iEEG[72,73] showing that channels even as far as 20 mm from the source can still record high-fidelity signals.

## Relative band power (RBP)

To evaluate the temporal dynamics of iEEG power activations, we used STFT to estimate the temporally resolved power spectral density (PSD), similar to our previous study[74]. Specifically, a temporal window, weighted by a Hann window to decrease the spectral leakage, was slid over raw iEEG data in time steps of 31.25 ms (corresponding to 16 samples at 512 Hz sampling rate). Note that the size of the STFT window was adjusted for the different frequency ranges (delta: 2-s long window; theta: 1-s long window; other bands: 0.5-s long window) to better estimate the power of the slower oscillations in the delta and theta bands. At each time step, the PSD was computed for each recording session yielding a PSD resolved time $t$ and frequency $f$ for each channel $ch$: PSD($t$, $f$, $ch$). The time $t$ corresponded to the center of the sliding window at a given time step.

As the choice of the STFT for time-frequency spectral estimation might be not unambiguous[75] and to ensure the validity of the STFT approach, we also applied the Morlet wavelet transform on the extracted trials, yielding another PSD estimate in the same format: PSD($t$, $f$, $ch$).

The PSD($t$, $f$, $ch$) was log-transformed to dB to make the distribution of spectral lines less skewed. To compensate for the $1/f$ power decay inherent to iEEG data[76], we normalized the spectra by computing the z-score over time for each frequency bin in each recording session:

$$PSD_{norm}(t,f,ch) = \frac{PSD(t,f,ch) - \mu(f,ch)}{\sigma(f,ch)}$$

where $PSD_{norm}$ is the normalized PSD, $\mu(f, ch)$ and $\sigma(f, ch)$ are the z-score normalization factors computed across the time of the entire recording session. The normalization factors, $\mu$ and $\sigma$, were robustly estimated, as the recording sessions lasted $205 \pm 14$ s (mean ± SEM across all recording sessions of all subjects).

To extract RBP, we divided the time-resolved power spectrum into six non-overlapping frequency bands ($fb$): delta (0.1–3 Hz), theta (4–7 Hz), alpha (8–12 Hz), beta (13–30 Hz), low-gamma band (31–48 Hz) and HGB (52–120 Hz). The notched out frequency ranges (48–52 Hz and 98–102 Hz) were not included into the HGB. Then, we averaged the normalized PSD within these bands. Finally, from each frequency band dataset, we extracted trials ($tr$), yielding a 4-D dataset of RBP($t$, $fb$, $ch$, $tr$). To obtain condition- and network-specific responses, the data were averaged across trials, based on the attention-switching condition $tr \in$ {external-to-internal, internal-to-external}, and across channels, based on channel localization to either the DMN or DAN, $ch \in$ {DMN, DAN}. In case of the RBP analysis with the rejected epochs (Suppl. Figs. S10b and S11b), we excluded the rejected epochs from the mean.

## Strength of the RBP difference

To quantify and compare the difference between the two conditions for each frequency, we computed a strength of RBP difference, which was calculated similar to a signal-to-noise ratio (SNR): here, the "signal" is the mean of each condition, and the "noise" is the trial-by-trial variance of each condition. For each channel $ch$, frequency band $fb$, and time point $t$, the strength of the RBP difference was defined as the variance of the condition means divided by mean trial-by-trial variances of each condition:

$$SNR(t,fb,ch) = \frac{var_c[mean_{tr}(RBP(t,fb,ch,tr_c))]}{mean_c[var_{tr}(RBP(t,fb,ch,tr_c))]}$$

where $tr_c$ are trials of attention-switching condition $c \in$ {external-to-internal, internal-to-external}. We computed the strength of the RBP difference as a log-transform of mean across the trial time from −2.3 s to 2.3 s, yielding SNR($fb$, $ch$) in dB.

## Timing of the RBP crossovers

Building on the observation of the activity crossover of the network RBP between the attention-switching conditions (Figs. 4d and 5d), we investigated the temporal order of crossover time points during attention switching separately for the DMN and DAN. The time of the crossover was defined for each significant channel and frequency band as the time point when the trial-averaged activity levels of the two attention-switching conditions crossed each other. Thus, for each attention-switching condition, we (1) subtracted the RBP activation common to both conditions (mean across all trials), and (2) low-pass filtered (1 Hz cutoff, 6th order Butterworth filter, zero phase shift) the trial-averaged RBP of each condition to smooth out the "jerkiness" in the data present even after trial averaging (see Suppl. Fig. S12). We further required that the crossover point was located in the time interval from −0.5 to 2.0 s after task switching to exclude potential outliers. In cases of multiple crossovers (back and forth), we considered only the first occurrence.

## Directed functional connectivity

The interaction between the networks was quantified by the directed functional connectivity (in particular, the DTF and PDC) using a graph analytic approach, in which the nodes of the network were sampled by iEEG channels defined by their locations, and the edges were connections between the nodes. The directed connectivity was obtained by fitting an MVAR model to the iEEG data:

$$X(t) = \sum_{l=1}^{p} A(l)X(t - l) + E(t)$$

where $X_t$ represents the vector of $k$ channels with raw iEEG signals at time $t$, $l$ denotes the time lag, $p$ is the model order (number of time lags in the MVAR model), $A_n$ is the $k$-by-$k$ matrix of autoregressive coefficients and $E_t$ is the channel—vector of the residual noise.

For each subject $s$, the input to the MVAR model consisted of trials extracted from the raw (or minimally preprocessed) data with selected channels localized in either the DMN or DAN: $D_s(t,ch,tr)$. All trials were z-scored in the temporal domain[61].

To increase the frequency specificity of the MVAR models, we distributed the DTF computation across four frequency bands: 0.1–30 Hz, 31–48 Hz, 52–98 Hz, 102–120 Hz (leaving out the 50 Hz line noise and its harmonics), similar to one of our previous study[77]. For each frequency band, the trials were band-pass (or low-pass) filtered (6th order Butterworth filter, zero phase shift). The order, $p$, of the MVAR models, which were fit separately in the four distinct frequency bands, can be determined, for example, by using the Akaike information criterion (AIC). In order to make the results easier to reproduce, we fixed the model order, which was equal to the bandwidth (e.g., $p = 17$ for the second band, as $48 - 31 = 17$). Note that we also reproduced the directed connectivity results by setting the model order to different values (e.g., used $p = 10$ for all bands or used the AIC, where $p = 9 \pm 1$, mean ± SEM, across all subjects), so the MVAR model order was not a critical parameter in the outcome of the directed functional connectivity analysis.

We selected three different measures to quantify the directed connectivity based on the MVAR model: the DTF[61] and PDC[78] and GGC[79].

$$DTF_{m,n}(f) = \sqrt{\frac{|H_{m,n}(f)|^2}{\sum_{j=1}^{k}|H_{m,j}(f)|^2}}$$

where $H_{m,n}(f)$ is a modeled transfer function between channels $m$ and $n$ at frequency $f$ and $k$ is the total number of iEEG channels. The strength of inter-node connection, in the interval [0,1], characterizes the proportion of information inflow from channel $n$ to channel $m$ related to total inflow to channel $m$. We selected the DTF from the multiple available methods of directed connectivity due to, for example, its low noise sensitivity[80]. The DTF

was also already successfully applied in some of our prior studies[77,81].

$$PDC_{m,n}(f) = \frac{|A_{m,n}(f)|}{\sqrt{A_n^*(f)\, A_n(f)}}$$

where $f$ represents frequency, $A_{m,n}(f)$ is an $m$th and $n$th element of Fourier-transformed MVAR model coefficients matrix $A(l)$, $A_n(f)$ is n-th column of matrix $A(f)$ and * denotes transpose and complex conjugate operator. The PDC also takes values from interval [0,1] and characterizes the outflow from channel $n$ to channel $m$ related to total outflow from the channel $n$[61].

$$GGC_{m,n}(f) = \frac{\left(Z_{n,n} - \frac{Z_{m,n}^2}{Z_{m,m}}\right)|H_{m,n}(f)|^2}{|S_{m,m}(f)|}$$

where $f$ represents frequency, $Z_{m,n}$ are elements of the covariance matrix $Z$ of the residual noise vector of the MVAR model for channels $m$ and $n$, $S_{m,m}$ is the power spectrum of channel $m$ based on the MVAR model and $H_{m,n}$ is the transfer function between $m$-th and $n$-th channels[79]. Note that here we used the MVAR model to compute the GGC from channel n to channel m, while in their original work Bressler et al. used a bivariate model[79].

To capture temporal changes in directed connectivity during attention switching, we computed the directed connectivity using a 500 ms–long sliding window in steps of 125 ms. At each time step, $t$, the MVAR model was computed across samples within the short window and all trials of each attention-switching condition $c$. Using the MVAR model, we estimated the DTF at the selected frequencies of the given frequency band, adopting the toolbox from Schlögl and Supp[82].

Iteration of this procedure across all four frequency bands and all time steps and conditions, produced a directed connectivity estimate (here exemplified for DTF, but similar notations hold also for PDC and GGC) between all channel pairs, i.e., a 5-D matrix for each subject $s$: $DTF_s(m,n,t,f,c)$, where $n$ is the source channel, $m$ is the target/sink channel, $t$ is time (corresponding to the middle of the sliding window), $f$ is a frequency bin and $c$ is the attention-switching condition. The DTF matrix was $z$-scored along the temporal dimension. To uncover the inter-network connectivity, we averaged the DTF for all source and target channels of each network (i.e., the DMN or DAN) separately for each subject $s$. The frequency-resolved DTF was then averaged into the six frequency bands $fb \in$ {delta, theta, …, HGB}—similar to RBP. Repeating this procedure for all subjects $s$ yielded $DTF_{DMN \to DAN}(t,fb,c,s)$ and $DTF_{DAN \to DMN}(t,fb,c,s)$.

### Phase-locking value (PLV)

To assess the functional connectivity between the channel pairs, we utilized the PLV[60], which is based on the assumption that inter-network communication is mediated by temporal synchronization of the instantaneous phase of their oscillations. To this end, we decomposed the iEEG signal into 2 Hz (non-overlapping) frequency bands $f \in$ (0–2–4–…–120 Hz) by applying a band-pass filter (Butterworth, 6th order, zero-phase shift). For each DMN-DAN channel pair, frequency band and trial, the time-resolved phase difference in these narrow bands was extracted by Hilbert transformation and the PLV computed as:

$$PLV(m,n,t,f,c) = \frac{1}{N}\left|\sum_{tr=1}^{N} e^{i[\varphi(t,f,m,tr) - \varphi(t,f,n,tr)]}\right|$$

where $\varphi$ is the instantaneous phase of the $m$th and $n$th channel at time $t$, $f$ denotes the narrow frequency band, $tr$ is trial and $N$ is the number of the trials of the attention switching condition $c$. The frequency-resolved PLV was then averaged into the six frequency bands $fb \in$ {delta, theta, …, HGB} —similar to RBP or the DTF.

### Statistics and reproducibility

To test for the significance of the difference in data distribution between the attention switching-conditions, we used the two-sided Wilcoxon rank sum test, with FDR correction for multiple testing[83], similar to our previous studies[84]. First, we assessed the significance of the difference in distribution between the attention switching conditions $c$ on a channel level, i.e. across the trials: $tr_c$, where $c \in$ {external-to-internal, internal-to-external}. Specifically, we conducted statistical tests for the RBP of each subject, RBP($t,fb,ch,tr_c$), for each time step $t$, frequency band $fb$ and channel $ch$. The significance level was $P = 0.05$, FDR corrected for multiple testing (across all subjects, time steps, frequency bands, and channels). Furthermore, we required that the significance of the difference was continuous for more than 100 ms to dampen false positive detections due to some instances of over-shooting in the noisy/jerky iEEG activity, similar to Tan et al.[35]. We refer to a channel satisfying these conditions simply as a "significant channel" (in a given frequency band).

Next, we tested for significance on a neural network level. The neural network activity in a given frequency band consisted of channels that met the above described significance criteria and were localized to either the DMN or DAN. The significance on a neural network level was tested across the channels comprising the network activity. Specifically, for the RBP of the network activity, RBP($t,fb,ch,c$), the significance of the difference between the attention-switching conditions $c$ was tested over the channels $ch$. The significance level was $P = 0.001$, FDR corrected for multiple testing (across networks, frequency bands, and time steps). We report not only the number of different channels ($N_C$) we tested in each network but also the number of subjects ($N_P$).

Finally, we report the significance of differences in network connectivity. We assessed the significance of differences between the attention-switching conditions $c$ for the DTF (or PDC, GGC, PLV) across subjects. Specifically, we tested over subjects for each time point $t$ in the $PLV_{DMN-DAN}(t, f, c, s)$, $DTF_{DMN \to DAN}(t, f, c, s)$ or $DTF_{DAN \to DMN}(t, f, c, s)$ (and similar to DTF also for PDC and GGC). The significance level was $P = 0.001$, FDR corrected for multiple testing (across time steps and frequencies).

### Reporting summary

Further information on research design is available in the Nature Portfolio Reporting Summary linked to this article.

### Data availability

The raw iEEG data that support the findings of this study as well as the code and numerical source for the figures are available in the Zenodo repository[85]. Further data are available from the corresponding authors upon reasonable request.

### Code availability

Code is publicly available for iEEG data analysis (https://github.com/JiriHammer/SEEG_dataAnalysis) and in the Zenodo repository[85].

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

## Acknowledgements

We would like to thank all patients for their participation. The research was supported by project nr. LX22NPO5107 (MEYS): Financed by European Union—Next Generation EU, the Czech Science Foundation (Grant No. 20-21339S), Grant Agency of Charles University (Grant No. 272221), and ERDF-Project Brain dynamics, No. CZ.02.01.01/00/22_008/0004643. Both Departments of Neurology and Pediatric Neurology, Second Faculty of Medicine, Charles University and Motol University Hospital are full members of ERN EpiCARE. All authors are members of the Epilepsy Research Centre Prague - EpiReC consortium.

## Author contributions

J.H.: Conceptualization, Methodology, Investigation, Formal analysis, Writing—original draft, Visualization, Funding acquisition. M.Ka.: Investigation, Data Curation, Visualization, Writing—review & editing. A.K.: Investigation, Data Curation, Writing—review & editing. D.K.: Investigation, Data Curation. P.F.: Investigation. M.Ku.: Investigation, Writing—review & editing. P.J.: Data Curation, Writing—review & editing. R.J.: Methodology, Formal analysis. P.K.: Investigation, Writing—review & editing. P.M.: Conceptualization, Resources, Writing—review & editing, Supervision, Funding acquisition.

## Competing interests

The authors declare no competing interests.
