## [Transparent Peer Review file · Communications Biology]

Antagonistic behavior of brain networks mediated by low-frequency oscillations: electrophysiological dynamics during internal–external attention switching

Corresponding Author: Dr Jiri Hammer

Figures originally included in the author's rebuttal have been redacted from this file.

Version 0:

Reviewer comments:

Reviewer #1

(Remarks to the Author)

Hammer et al., undertake intracranial EEG (iEEG) analyses, obtained from 25 patients undergoing surgical evaluation for epilepsy, to investigate the antagonistic activity between default mode network (DMN) and dorsal attention network (DAN). Prior literature has demonstrated such antagonistic activity of networks and similar evidence from iEEG data is scarce. The authors use a task paradigm with internal and external attention required to tease out these network effects. The authors do find such antagonistic activity/attention switching in low frequency bands of theta, alpha and higher frequency band. Further the low frequency activity was concomitant with increase in high frequency activity. Only a couple of metrics of relative band power, phase locking value and directed transfer function are used for analytical quantification. The study has interesting objective and aims and is clear and the sample size is also optimal especially for data from neurosurgical patients. The data is unique. I have some major concerns about the analytical strategies applied and these unfortunately make me question the results. I elaborate on it below apart from some other comments

1) The data was high pass filtered at .1 Hz, yet the frequency bands later employed for delta is 0-3Hz. Similarly, the notch filtering employed at 50, 100 and 150 Hz with +2 Hz of frequency range notching, yet the low gamma band 31-50 Hz and high gamma band taken as 51-120Hz. This obviously introduces spurious signal and thus questionable results

2)The authors use 500ms long window to compute the temporal dynamics of power modulation. However, 500ms would only resolve accurate power estimation for 2Hz and above only. Thus, the delta frequency power as computed from 0-3 Hz would not be reliable.

3) The authors mention in the methods section of "Relative Band power" the 1/f power decay was compensated by whitening the spectra with zscore computation over time over each frequency band. I am not sure how this technique compensates for 1/f power. Further details of the methods are required. Zscoring for each frequency bin would scale the data according to the frequency amplitudes inherent in the data and the noise which might skew the results.

4) The authors also mention that no trials were rejected in the Trial extraction section. I am surprised that no trials were rejected. iEEG data from patients with epilepsy will have inherent artifacts interictal or others and even though channels from seizure onset or irritative zone were excluded, such spike or interictal discharges generate network activity affecting other non-seizure related channels. Visual data inspection is perhaps one of the most important steps in iEEG analyses and that no trials were rejected from any patients is very surprising.

5) The authors use band limited power in the frequency bands and then move to use directed transfer function(DTF) and phase locking value (PLV) to measure connectivity. However, given the myriad methods of connectivity specific justifications and motivations are missing.

6)The authors mention DTF as effective connectivity. However, I would be cautious in interpreting DTF as effective connectivity as it essentially is directed connectivity based on temporal precedence like granger causality (see Chiarion et al., Bioengineering 10, no. 3 2023; Friston Brain Connectivity 2011; Friston, Moran & Seth 2013)

Reviewer #2

(Remarks to the Author)

The authors studied the EEG frequency activity during attention switching between internal and external modes of information processing in two antagonistic brain networks: the DMN and DAN. They used a novel paradigm to study this topic and also analyzed in detail the functional and effective connectivity between the two brain networks. I was impressed by the thorough analytical approach of this work. Most of the results are very straightforward. They will be interest to the big community that is interested in large-scale correlations in the brain, and especially between intrinsic functional connectivity networks. I particularly appreciated the very clear way in which all analytical methods were described in the Methods section. I only have a number of relatively minor comments, most regarding clarification.

- 1) There is quite some redundancy between the second and third paragraph of the Introduction. That needs to be fixed.
- 2) In line 84, a new paragraph starts with “Most of the evidence for the anticorrelated, antagonistic activity of the DMN ...”, which does not mention the DAN. This was a bit surprising, given the preceding paragraph. Is this paragraph really only about the DMN?
- 3) In the Figure 1 caption, the authors refer to the ‘alternating’ tasks. But ‘alternating’ means ABABABAB (i.e. switching on each trial), which does not seem to correspond with Figure 1c. In the Methods section, the authors could also mention that the task did not switch on every trial.
- 4) Section 2.3, paragraph “Rich spectral power dynamics ...”. I couldn’t quite follow the text here. This is in part because I did not understand what the authors meant with ‘late’, and because the authors refer to ‘I-task’ and ‘E-task’, but do not clarify whether they refer to the task before or after the switch ($t=0$). I would appreciate if the authors make an effort to make this paragraph crystal clear in terms of what we should look at in Figure 3.
- 5) The authors use a LOT of acronyms, which is not in the interest of the reader. Are all of these acronyms necessary?
- 6) Discussion, paragraph “Using the fast temporal resolution of the iEEG ...”. Can the authors make explicit what their study adds to these previous studies in terms of methodology?
- 7) “Notably, our results suggest the possibility that the interactions between the DMN 539 and DAN are mediated by another network.” I did not understand why.
- 8) Line 579. Report SD (i.e. a descriptive statistic) instead of SEM in the context of subject characteristics.
- 9) Line 614: change “The entire paradigm” to “The entire experiment”?

Reviewer #3

(Remarks to the Author)

In this paper, the authors report the results of an experiment they were able to perform using iEEG electrodes implanted for clinical reasons. The experiment consisted of a series of two consecutive tasks which required the participants to switch from internal to external processing or the other way around. iEEG electrodes were located in either the Dorsal Attention Network, or the Default Mode Network. The data is analyzed per network. The DMN shows a switch from low-frequency to high-frequency content when switching from the external to the internal task, whereas the opposite is true in the DAN.

This suggests that during the internal task the DMN is activated with the DAN being inactivated (and vv). I am wondering what this means for the rest of the neurophysiological nature where we mainly tap in the lower frequencies: does this mean that EEG activations in the lower frequency band indicate an inactive network and that we simply cannot access the real activations hidden in the high gamma band?

The paper is well-written and presents an interesting topic. I have some methodological questions that I would like to have clarified

- The first remark is that I do not understand why the authors use the STFT instead of a wavelet based approach. A wavelet approach provides a better time resolution at higher frequencies and better frequency resolution at low frequencies, and both aspects seem to be important for the analyses presented in this paper. For example, a 500 ms-long time window is a rather large time window to pick up changes across time and still has a relatively poor frequency resolution ($\Delta f = 2\text{Hz}$). The following link could be helpful: <https://arxiv.org/pdf/2101.06707>

- I am unsure what the authors meant by “To compensate for the $1/f$ power decay inherent to iEEG data, we whitened the spectra by computing the z-score over time for each frequency bin in each recording session”. I only use the term whitening to refer to the process of rotating the signals in such a way that the covariance matrix becomes the identity matrix (https://en.wikipedia.org/wiki/Whitening_transformation). I assume what the authors mean is that the signal was normalized per frequency bin? Is this correct? $z(t,f,ch) = (\text{PSD}(t,f,ch) - \text{mean}(\text{PSD}(:,f,ch),1)) / \text{std}(\text{PSD}(:,f,ch),1)$? A bit more mathematical details would help me understand what exactly has been done

- a few lines below that, the authors state "Then, we averaged the normalized PSD". Again, this is confusing as the authors have mentioned whitening but not normalisation. Do the authors refer to normalisation across the spectrum, e.g. as often done as follows: $\text{normalized_PSD}(f) = \text{PSD}(f) / \sum(\text{PSD}(:))$? If this is the case, the changes observed in the paper are not independent: an increase in the higher frequencies would automatically result in a decrease in lower frequencies. This should be clarified in the Methods section.

- the colour scale in Figure 3 for the z-score of the PSD is a bit odd. Why do the authors use a cutoff of 0.05? This results in relatively large dark blobs. Couldn't it be that the response is more specific to a specific frequency and that this is masked? In addition, I would suggest to try a logarithmic transformation of the frequency axis as this would free up space for the lower frequencies and reduce the amount of space used for the higher frequencies (where everything seems to be similar anyway).

With respect to the results, I have the following questions:

- how does the spectrum look like for the external and internal tasks in the respective networks (so Figures 3b and 3d but without the time variations)

- why do the authors bother to divide the low-frequencies into the typical frequency bands? From the results it seems to be the case that all low-frequencies are suppressed (or activated) vs the high-frequencies. In order to understand this general suppression of all low frequencies, it would be important to know whether the normalisation has been performed across frequencies (see previous question).

- if no normalisation of the spectrum is done, how would the authors explain that all low frequency bands react similarly (Figure 5) despite their strongly diverging functions in human brain functioning?

- I don't understand why the authors state that task switching occurs at 0.6s (line 243). The task is switched at $t=0$ (the moment of the BP), no?

Author Rebuttal letter:

Reviewers' comments:

Reviewer #1 (Remarks to the Author):

Hammer et al., undertake intracranial EEG (iEEG) analyses, obtained from 25 patients undergoing surgical evaluation for epilepsy, to investigate the antagonistic activity between default mode network (DMN) and dorsal attention network (DAN). Prior literature has demonstrated such antagonistic activity of networks and similar evidence from iEEG data is scarce. The authors use a task paradigm with internal and external attention required to tease out these network effects. The authors do find such antagonistic activity/attention switching in low frequency bands of theta, alpha and higher frequency band. Further the low frequency activity was concomitant with increase in high frequency activity. Only a couple of metrics of relative band power, phase locking value and directed transfer function are used for analytical quantification. The study has interesting objective and aims and is clear and the sample size is also optimal especially for data from neurosurgical patients. The data is unique. I have some major concerns about the analytical strategies applied and these unfortunately make me question the results. I elaborate on it below apart from some other comments

We would like to thank the reviewer for her or his thoughtful comments, which led to substantial methodological improvements. Below is our point-by-point reply to the comments.

1) The data was high pass filtered at .1 Hz, yet the frequency bands later employed for delta is 0-3Hz. Similarly, the notch filtering employed at 50, 100 and 150 Hz with +-2 Hz of frequency range notching, yet the low gamma band 31-50 Hz and high gamma band taken as 51-120Hz. This obviously introduces spurious signal and thus questionable results

We would like to thank the reviewer for picking up this inconsistency. We corrected the frequency range for the delta band to 0.1–3 Hz, low-gamma to 31–48 Hz and high-gamma to 52–120 Hz in the manuscript text and figures and explained that the notched out frequencies were not included in the relative band power of the HGB.

In more detail, the changes in the manuscript include:

Methods part (see lines: 776–779): "To extract RBP, we divided the time-resolved power spectrum into six non-overlapping frequency bands (fb): delta (0.1–3 Hz), theta (4–7 Hz), alpha (8–12 Hz), beta (13–30 Hz), low-gamma band (31–50 48 Hz) and HGB (52–120 Hz). The notched out frequency ranges (48–52 Hz and 98–102 Hz) were not included into the HGB."

Results part (see lines: 208–209): "delta (0.1–3 Hz), theta (4–7 Hz), alpha (8–12 Hz), beta (13–30 Hz), the low-gamma band (LGB, 31–48 Hz) and the HGB (52–120 Hz)."

2) The authors use 500ms long window to compute the temporal dynamics of power modulation. However, 500ms would only resolve accurate power estimation for 2Hz and above only. Thus, the delta frequency power as computed from 0-3 Hz would not be reliable.

This is indeed a very good point. We changed the window size for the low-frequency bands: delta to 2-s long window and theta to 1-s long window. We updated the corresponding Methods section accordingly as well as the Figs. 4–7, and Suppl. Figs. S6–S11, S13. The changes in the window size led to improvements in the delta and theta power estimation, reflected in the strength of the RBP difference during attention switching (Results section 2.6).

In more detail, the changes in the manuscript include:

Methods (see lines: 757–759): “Note that the size of the STFT window was adjusted for the different frequency ranges (delta: 2-s long window; theta: 1-s long window; other bands: 0.5-s long window) to better estimate the power of the slower oscillations in the delta and theta bands.”

Results (see lines: 344–349): “For the DMN, the strength of the RBP difference between both conditions was highest in the delta, theta and theta alpha bands. The differences in these low-frequency bands were significantly stronger than those in the HGB ($P < 0.001$, two-tailed Wilcoxon rank sum test). For the DAN, the most pronounced differences were also found in the low-frequency bands (alpha, delta and theta) band, albeit not significantly larger than in the HGB ($P > 0.001$).”

We also list the updated figures (a recommendation from Communications Biology) and indicate the changes:

3) The authors mention in the methods section of “Relative Band power” the $1/f$ power decay was compensated by whitening the spectra with zscore computation over time over each frequency band. I am not sure how this technique compensates for $1/f$ power. Further details of the methods are required. Zscoring for each frequency bin would scale the data according to the frequency amplitudes inherent in the data and the noise which might skew the results.

We included a formula to make it clear how the spectral whitening normalization was performed (see lines: 714–719). The argument that the z-scoring for each frequency bin may skew the results because of noise is certainly of concern. However, we are convinced that our estimate of the z-score factors (mean and SD) is quite robust, as it was computed across the entire recording session lasting 205 ± 14 s (mean \pm SEM across subjects). We added a note on this issue in the manuscript. Moreover, we did not observe almost any changes in the estimated spectral power after rejection of time epochs of epileptic events, such as interictal discharges (see Suppl. Figs. S10b and S11b and also our response to the comment below).

In more detail, the changes in the manuscript include:

Methods (see lines 771–775): “

where PSD_{norm} is the normalized PSD, $\mu(f, ch)$ and $\sigma(f, ch)$ are the z-score normalization factors computed across the time of the entire recording session. The normalization factors, μ and σ , were robustly estimated, as the recording sessions lasted 205 ± 14 s (mean \pm SEM across all recording sessions of all subjects).”

4) The authors also mention that no trials were rejected in the Trial extraction section. I am surprised that no trials were rejected. iEEG data from patients with epilepsy will have inherent artifacts interictal or others and even though channels from seizure onset or irritative zone were excluded, such spike or interictal discharges generate network activity affecting other non-seizure related channels. Visual data inspection is perhaps one of the most important steps in iEEG analyses and that no trials were rejected from any patients is very surprising.

We agree with the reviewer that the visual inspection of the data is indeed one of the most important steps in iEEG data analysis. In fact, before the initial submission of our manuscript, we performed our analyses both with and without data rejection. As we did not observe any major changes in the results, we decided to report on the easier version: without data rejection. Thus, although there could be at times a contamination with epilepsy- (or other-) related activity, these “artifacts” acted as an additional and, importantly,

task-unrelated noise, which was attenuated (averaged out) in the reported RBP of network activity.

However, based on this comment, which is a very good one and likely to be shared by the majority of the readers, we decided to include supplementary figures of RBP activations with data rejection in the revised version of the manuscript (see Suppl. Figs. S10b and S11b). We opted for a published algorithm for epileptic spike detection, used in some of our previous studies as well. The advantage of an algorithmic solution for epileptic spike detection (followed by visual inspection) is that it is reproducible and also reusable for other studies. While in the RBP analysis the rejection of epileptic events can be done for each iEEG channel separately, the data rejection for the directed connectivity analysis based on MVAR model is more problematic, because the MVAR model uses the channels altogether and the likelihood that a trial (or a windowed epoch of the trial) will be rejected adds up, resulting in the end in not enough data or unbalanced trial numbers for each condition. For these reasons, we decided not to include the trial rejection analysis in the main text of the manuscript, but rather as supplementary figures to assure the readers about the validity of the results.

In more detail, the changes in the manuscript include:

Methods (see lines: 720–732): “In the presented results, we made no trial rejection.

However, a rightful concern could arise to which degree the results could be affected by various artifacts, mostly from the epileptic activity (such as inter-ictal discharges). Hence, we replicated the analyses of RBP activations including data rejection (Suppl. Fig. S10b and S11b). In particular, we rejected epochs of epileptic activity (e.g., inter-ictal discharges) as the epileptic network activity may affect the other, non-seizure related channels as well. To this end, we opted for an epileptic spike detector⁸⁰, used in our previous studies^{64,81,82}. Additionally, we also rejected those epochs, where the amplitude of the raw iEEG data was above or below six SDs from the mean (defined for each channel and recording session separately). The indices of the detected epochs (of individual duration) with additional, 100-ms long margins were rejected from the analyses, resulting in 0.6 ± 0.1 % of rejected data from the trials; mean \pm SEM across 25 subjects). Note that we did not reject whole trials, in order to preserve maximum of the data and trials, but only epochs of specific duration.”

Methods (see lines: 785–786): “In case of the RBP analyzes with the rejected epochs (Suppl. Fig. S10b and S11b), we excluded the rejected epochs from the mean.”

Results (see lines: 351–356): “Note that in the above analyses, we did not perform any trial or epoch rejection. To mitigate possible concerns about the role of artifacts in the iEEG data, we repeated the RBP analysis including the data rejection (see Methods for more details). The RBP activations of the DMN (Suppl. Fig. S10b) and DAN (Suppl. Fig. S11b) were almost identical, which can be explained by the large number of trials and channels comprising the network-level activity and the rather low susceptibility of iEEG to artifacts in general⁷⁷.”

5) The authors use band limited power in the frequency bands and then move to use directed transfer function (DTF) and phase locking value (PLV) to measure connectivity. However, given the myriad methods of connectivity specific justifications and motivations are missing.

We added a motivation for the choice of the functional connectivity measures, which was—admittedly—missing (see lines: 363-373): “We selected two different, data-driven and commonly used approaches: a non-directed connectivity measure, the phase-locking value (PLV), and a directed connectivity based on multivariate autoregressive (MVAR) model: the directed transfer function (DTF), the partial directed coherence (PDC) and the Granger-Geweke causality (GGC). Given the large number of connectivity estimators, one of the reasons for selecting this particular set lied in their different characteristics: the PLV is solely based on phase synchronization of iEEG signals, while the directed connectivity measures based on the MVAR model take into account both phase and amplitude of the signals.”

Additionally, to increase the number of the different connectivity measures, we also included other measures of the directional connectivity, namely the partial directed coherence (see Suppl. Fig. S14) and Granger-Geweke causality (see Suppl. Fig. S15).

In more detail, the changes in the manuscript include:

Updated and restructured the Methods section accordingly (see lines: 800–863).

Results (see lines: 436–437): “Similar results were observed for the PDC (Suppl. Fig. S14) and GGC (Suppl. Fig. S15).” and additional small changes in line 425.

Discussion (see lines: 591–594): “The high similarity among the directed connectivity measures (DTF, PDC and GGC) can be likely attributed to averaging across the different channels of both networks in each subject and also to the fact that they were computed using the same MVAR model.”

6) The authors mention DTF as effective connectivity. However, I would be cautious in interpreting DTF as effective connectivity as it essentially is directed connectivity based on temporal precedence like granger causality (see Chiarion et al.,

Bioengineering 10, no. 3 2023; Friston Brain Connectivity 2011; Friston, Moran & Seth 2013)

We would like to thank the reviewer for raising this point. The literature is unfortunately not yet very consistent in the use of “effective connectivity”. In some studies (very much like in our initial submission), any directed connectivity measure is referred to as the effective connectivity. From the studies that the reviewer kindly pointed to us follows that effective connectivity requires an explicit, mechanistic model of cause–effect relationship, which was not the case in our study.

We have thus adopted the following changes: (1) replaced the term “effective connectivity” by “directed (functional) connectivity” (see multiple changes in the manuscript), (2) explicitly mentioned the fact that our functional connectivity was “data driven” (see line: 397) and provided the useful citations of Chiaron et al. (2023) and Friston (2011) for further information for interested readers (see line: 560).

Reviewer #2 (Remarks to the Author):

The authors studied the EEG frequency activity during attention switching between internal and external modes of information processing in two antagonistic brain networks: the DMN and DAN. They used a novel paradigm to study this topic and also analyzed in detail the functional and effective connectivity between the two brain networks. I was impressed by the thorough analytical approach of this work. Most of the results are very straightforward. They will be interest to the big community that is interested in large-scale correlations in the brain, and especially between intrinsic functional connectivity networks. I particularly appreciated the very clear way in which all analytical methods were described in the Methods section. I only have a number of relatively minor comments, most regarding clarification.

We would like to thank the reviewer for her or his thoughtful comments leading to improvements in the manuscript. Below is our point-by-point reply to the comments.

1) There is quite some redundancy between the second and third paragraph of the Introduction. That needs to be fixed.

Indeed, the two paragraphs overlapped and we would like to thank the reviewer for this observation. We deleted the redundant (repeating) information from the third paragraph and merged the third paragraph with the fourth paragraph (this now-united paragraph is meant to provide readers with a quick introduction to the concept of the DMN and further relevant references).

In more detail, the changes in the manuscript include (see lines: 59–69):

“The DMN, initially discovered as the resting state “default” (or “task-negative”) network^{4,5,7,9} (for recent reviews see^{12–15}), is a large-scale neural network distributed over the association cortex, comprising areas in the frontal (medial and anterior prefrontal cortex), temporal (lateral temporal cortex and medial temporal lobe), and parietal (posterior cingulate cortex and inferior parietal lobule) lobes. Later, the DMN was shown not only to be the task-negative network, but also ...”.

2) In line 84, a new paragraph starts with “Most of the evidence for the anticorrelated, antagonistic activity of the DMN ...”, which does not mention the DAN. This was a bit surprising, given the preceding paragraph. Is this paragraph really only about the DMN?

The paragraph was not meant to report exclusively on the DMN, but rather on the anticorrelations of the DMN with other large-scale brain networks. We agree with the reviewer that our formulation could be confusing and modified the first sentence: “Most of the evidence for the anticorrelated, antagonistic activity of the large-scale brain networks, such as the DMN and DAN, ...” (see lines: 85–86).

3) In the Figure 1 caption, the authors refer to the ‘alternating’ tasks. But ‘alternating’ means ABABABAB (i.e. switching on each trial), which does not seem to correspond with Figure 1c. In the Methods section, the authors could also mention that the task did not switch on every trial.

We would like to thank the reviewer for picking up this inconsistency, which could confuse the readers. Indeed, the tasks were not alternating. In the Figure 1 caption, we replaced the word “alternating” with “successive” (see line: 153). We also added a note in the Methods section (see line: 671): “Note that the task did not switch on every trial”.

4) Section 2.3, paragraph “Rich spectral power dynamics ...”. I couldn’t quite follow the text here. This is in part because I did not understand what the authors meant with

'late', and because the authors refer to 'I-task' and 'E-task', but do not clarify whether they refer to the task before or after the switch ($t=0$). I would appreciate if the authors make an effort to make this paragraph crystal clear in terms of what we should look at in Figure 3.

We agree with the reviewer that this paragraph was indeed hard to follow and would like to thank for raising our attention to it. We decided to delete this paragraph altogether (lines: 197–205), as the dynamics of both networks is anyhow detailed below (DMN: Section 2.4, Fig. 4; DAN: Section 2.5, Fig. 5), containing the same information, but easier to follow and presented clearly with frequency-specific graphs. This deletion has neither any impact on the information content presented in the manuscript, nor on the text flow. The main message of this paragraph remains unchanged: to show that the single-channel example shown in the Section 2.2 and Fig. 2 was not "just an exception", but can be robustly reproduced on the network activity level.

5) The authors use a LOT of acronyms, which is not in the interest of the reader. Are all of these acronyms necessary?

This is a very good point, as acronyms facilitate writing, but not necessarily reading of the text. In line with the reviewers comment, we decided to remove the acronyms, which were highly specific for our text and are not commonly found in the literature. In particular, we replaced: LFB by "low-frequency band", LGB by "low-gamma band", E-I by "external-to-internal", I-E by "internal-to-external", E-task to "external attention task", I-task to "internal attention task". See multiple changes in the manuscript. The other acronyms (iEEG, HGB, DMN, DAN, DTF, PLV, STFT, PSD, MVAR, ...) have been quite commonly used in previous studies as well and, hence, the readers may be well acquainted with them.

6) Discussion, paragraph "Using the fast temporal resolution of the iEEG ...". Can the authors make explicit what their study adds to these previous studies in terms of methodology?

We made explicit that our methodology was different, focusing on the crossover points of neuronal activity during attention switching than the activation onset/peak reported in the studies mentioned in that paragraph (see lines: 546–548): "While these studies investigated activation onsets (or peaks), our methodology was different, as we focused on the crossover points of neuronal activity during attention switching."

We also believe that now the transition to the next paragraph is more smooth and we would like to thank the reviewer for raising this point.

7) "Notably, our results suggest the possibility that the interactions between the DMN 539 and DAN are mediated by another network." I did not understand why.

A very good point, as this was an unfortunate formulation. Indeed, our results do not "suggest" this. We changed it to a better formulation: "our results do not exclude the possibility ..." (see line: 602).

8) Line 579. Report SD (i.e. a descriptive statistic) instead of SEM in the context of subject characteristics.

Corrected (see line: 643).

9) Line 614: change "The entire paradigm" to "The entire experiment"?

Corrected (see line: 681).

Reviewer #3 (Remarks to the Author):

In this paper, the authors report the results of an experiment they were able to perform using iEEG electrodes implanted for clinical reasons. The experiment consisted of a series of two consecutive tasks which required the participants to switch from internal to external processing or the other way around. iEEG electrodes were located in either the Dorsal Attention Network, or the Default Mode Network. The data is analyzed per network. The DMN shows a switch from low-frequency to high-frequency content when switching from the external to the internal task, whereas the opposite is true in the DAN.

We would like to thank the reviewer for her or his thoughtful comments allowing us to further improve the manuscript. Please, see our point-by-point reply below.

1) This suggests that during the internal task the DMN is activated with the DAN being

inactivated (and vv). I am wondering what this means for the rest of the neurophysiological nature where we mainly tap in the lower frequencies: does this mean that EEG activations in the lower frequency band indicate an inactive network and that we simply cannot access the real activations hidden in the high gamma band?

This is quite a complex question and we are afraid we cannot provide an easy answer. When examining the extracellular voltage power spectrum, the shift from rest to task is often associated with an increase in power at high frequencies (> 50 Hz), and often accompanied by a decrease in power at low frequencies (< 30 Hz) (e.g., Crone et al., 1998), which was observed in a number of cognitive tasks and different brain regions (e.g., Miller et al., 2014). Specifically, the alpha band (8–12 Hz) oscillations were suggested to be the 'idling' rhythm (Pfurtscheller et al., 1996; Klimesch et al., 2007; Jensen & Mazaheri, 2010), the beta band was hypothesized to signal the maintenance (a status-quo) of the current state (Engel & Fries, 2010). Our study would be, generally, in line with these observations. However, to state that "the lower frequency band indicate an inactive network" is perhaps a bit too general. The situation is more complex, as the high-frequency oscillations (such as high-gamma) can be phase-locked to low-frequency (such as delta, theta bands) oscillations (Fries, 2015) and serve as a mechanism to transfer information across large-scale brain networks (Canolty & Knight, 2010). For example, it was shown that when the hippocampus communicates with target regions, the theta power in the two regions becomes high (Lisman & Jensen, 2013; Herweg et al., 2020). In short, the activations/deactivations in the low-frequency bands should be interpreted with caution.

Citations:

- Crone, N. E., Miglioretti, D. L., Gordon, B., Sieracki, J. M., Wilson, M. T., Uematsu, S., & Lesser, R. P. (1998). Functional mapping of human sensorimotor cortex with electrocorticographic spectral analysis. I. Alpha and beta event-related desynchronization. *Brain: a journal of neurology*, 121(12), 2271-2299.
- Miller, K. J., Honey, C. J., Hermes, D., Rao, R. P., & Ojemann, J. G. (2014). Broadband changes in the cortical surface potential track activation of functionally diverse neuronal populations. *Neuroimage*, 85, 711-720.
- Pfurtscheller, G., Stancak Jr, A., & Neuper, C. (1996). Event-related synchronization (ERS) in the alpha band—an electrophysiological correlate of cortical idling: a review. *International journal of psychophysiology*, 24(1-2), 39-46.
- Klimesch, W., Sauseng, P., & Hanslmayr, S. (2007). EEG alpha oscillations: the inhibition–timing hypothesis. *Brain research reviews*, 53(1), 63-88.
- Jensen, O., & Mazaheri, A. (2010). Shaping functional architecture by oscillatory alpha activity: gating by inhibition. *Frontiers in human neuroscience*, 4, 186.
- Engel, A. K., & Fries, P. (2010). Beta-band oscillations—signalling the status quo?. *Current opinion in neurobiology*, 20(2), 156-165.
- Fries, P. (2015). Rhythms for cognition: communication through coherence. *Neuron*, 88(1), 220-235.
- Canolty, R. T., & Knight, R. T. (2010). The functional role of cross-frequency coupling. *Trends in cognitive sciences*, 14(11), 506-515.
- Lisman, J. E., & Jensen, O. (2013). The theta-gamma neural code. *Neuron*, 77(6), 1002-1016.
- Herweg, N. A., Solomon, E. A., & Kahana, M. J. (2020). Theta oscillations in human memory. *Trends in cognitive sciences*, 24(3), 208-227.

The paper is well-written and presents an interesting topic. I have some methodological questions that I would like to have clarified

2) - The first remark is that I do not understand why the authors use the STFT instead of a wavelet based approach. A wavelet approach provides a better time resolution at higher frequencies and better frequency resolution at low frequencies, and both aspects seem to be important for the analyses presented in this paper. For example, a 500 ms-long time window is a rather large time window to pick up changes across time and still has a relatively poor frequency resolution ($\Delta f = 2\text{Hz}$). The following link could be helpful: <https://arxiv.org/pdf/2101.06707>

We would like to thank the reviewer for raising this issue, which is likely to be shared by many other readers as well. Our choice of STFT for time-frequency decomposition of iEEG signals was guided (somewhat practically) by its successful application in some of our previous studies. A wavelet approach is certainly usable as well. We compared the relative band power using the STFT and the Morlet wavelet transform. Both methods converged to a very similar solution, when computed on the same datasets.

We updated the manuscript in the:

Results section (see lines: 356–362): "To ensure that the RBP was correctly (albeit not necessarily optimally) estimated by the STFT with variable window size, we repeated the

RBP analysis using the Morlet wavelet transformation (see Methods for more details), leading again to very similar results (correlation coefficient between the RBP estimated by STFT and the wavelet transform was 0.90 ± 0.05 for DMN and 0.93 ± 0.03 for DAN; mean \pm SEM across all frequency bands and conditions; Suppl. Fig. S10c and S11c)."

Methods (see lines: 763–766, including the useful citation provided by the reviewer for readers interested in further details): "As the choice of the STFT for time-frequency spectral estimation might be not unambiguous⁸³ and to ensure the validity of the STFT approach, we also applied the Morlet wavelet transform on the extracted trials, yielding another PSD estimate in the same format: PSD(t,f,ch)."

Supplementary figures S4c and S5c.

To avoid a potentially false impression that the STFT method is superior to other methods, we explicitly state that the choice of the STFT for RBP estimation might be "not necessarily optimal".

3) - I am unsure what the authors meant by "To compensate for the 1/f power decay inherent to iEEG data, we whitened the spectra by computing the z-score over time for each frequency bin in each recording session". I only use the term whitening to refer to the process of rotating the signals in such a way that the covariance matrix becomes the identity matrix (https://en.wikipedia.org/wiki/Whitening_transformation). I assume what the authors mean is that the signal was normalized per frequency bin? Is this correct? $z(t,f,ch) = (\text{PSD}(t,f,ch) - \text{mean}(\text{PSD}(:,f,ch),1)) / \text{std}(\text{PSD}(:,f,ch),1)$? A bit more mathematical details would help me understand what exactly has been done. To address the reviewer's concern about the "whitening transformation" (which indeed can be used in a different context), we avoided the term "whitening" and replaced it with "normalization" and a consistent usage of the term "normalized PSD". We also provide the formula for the spectral normalization via z-score computation for each frequency bin, which was done exactly as the reviewer anticipated, but which was indeed missing in the original version of the manuscript (see lines: 771–775): where PSDnorm is the normalized PSD, $\mu(f,ch)$ and $\sigma(f,ch)$ are the z-score normalization factors computed across the time of the entire recording session. The normalization factors, μ and σ , were robustly estimated, as the recording sessions lasted 205 ± 14 s (mean \pm SEM across all recording sessions of all subjects).

4) - a few lines below that, the authors state "Then, we averaged the normalized PSD". Again, this is confusing as the authors have mentioned whitening but not normalisation. Do the authors refer to normalisation across the spectrum, e.g. as often done as follows: $\text{normalized_PSD}(f) = \text{PSD}(f) / \text{sum}(\text{PSD}(:))$? If this is the case, the changes observed in the paper are not independent: an increase in the higher frequencies would automatically result in a decrease in lower frequencies. This should be clarified in the Methods section.

No additional normalization (e.g., across frequency bins of the spectrograms) was performed. This confusion was likely caused by our unfortunate and especially inconsistent use of the term "normalized PSD". In the revised version, the "normalized PSD" is used consistently and we also provide a formula for this normalization, as explained above.

5) - the colour scale in Figure 3 for the z-score of the PSD is a bit odd. Why do the authors use a cutoff of 0.05? This results in relatively large dark blobs. Couldn't it be that the response is more specific to a specific frequency and that this is masked? In addition, I would suggest to try a logarithmic transformation of the frequency axis as this would free up space for the lower frequencies and reduce the amount of space used for the higher frequencies (where everything seems to be similar anyway). In the initial version of the manuscript, we used the lower cutoff with the intention to highlight the changes in the high-frequency power. The downside of this were the "relatively large dark blobs" - as the reviewer correctly pointed out. Based on this comment, we revised the figure, enlarged the color-scale to avoid the dark blobs with color-code saturation and modified the y-axis to a logarithmic scale, as suggested by the reviewer. See the corresponding changes in Fig. 3 and the figure legend (line: 218–219).

With respect to the results, I have the following questions:

6) - how does the spectrum look like for the external and internal tasks in the respective networks (so Figures 3b and 3d but without the time variations)
This is a good question and we were happy to provide the figure (see Suppl. Fig. 3). As we believe that this could be of interest for many readers, we decided to include the figure into the Supplementary material. See the corresponding changes in the manuscript

(lines: 195–196): “PSD for the external and internal attention tasks of both networks can be found in Suppl. Fig. S3.”

7) - why do the authors bother to divide the low-frequencies into the typical frequency bands? From the results it seems to be the case that all low-frequencies are suppressed (or activated) vs the high-frequencies. In order to understand this general suppression of all low frequencies, it would be important to know whether the normalisation has been performed across frequencies (see previous question). As explained above, we did not perform the normalization of the spectrograms across frequencies. However, the low-frequency bands (< 30 Hz) indeed show very similar activations. In fact, during the process of writing the manuscript, we even had a version, where we put the low-frequencies into one “low-frequency band (0–30 Hz)”, exactly because of the similarity in their activations. However, the results from functional connectivity (different PLV and DTF in the delta, theta, alpha and beta bands) convinced us to divide the spectrum into the typical frequency bands.

8) - if no normalisation of the spectrum is done, how would the authors explain that all low frequency bands react similarly (Figure 5) despite their strongly diverging functions in human brain functioning?

Good question, but—again—not easy to answer. To be honest, we were also puzzled that the low-frequency bands (< 30 Hz) reacted so similarly. However, despite their similar temporal profile of RBP, the different low-frequency bands still could play different functional roles. For example, in the phase-amplitude coupling between low-frequency phase (in the delta, theta bands) and the gamma band (> 30 Hz) amplitude, or in gating by inhibition in alpha oscillations. Clearly, more research is needed to elucidate whether (and to what extent) the different frequency bands are independent, as well as their exact functional role. However, to tackle these issues in detail would be, in our opinion, beyond the scope of the current manuscript. Note that in the Discussion of the initially submitted manuscript we pointed out this issue by stating: “A future study could address the phase-amplitude coupling for possible nested oscillations among the different frequency bands, for example, clarifying whether the inter-network communication is established by phase synchronization of oscillations at lower frequencies, acting as a temporal reference frame for information carried by high-frequency activity⁶⁰” (see lines: 597–601).

9) - I don't understand why the authors state that task switching occurs at 0.6s (line 243). The task is switched at $t=0$ (the moment of the BP), no?

Right, the switching occurred at $t = 0$ s. This was our rather unfortunate formulation. Based on this comment (see also the comment 4 of reviewer 2), we decided to delete this paragraph (see lines: 197–205), because a detailed description of the activations is provided in the Sections 2.4 and 2.5 (together with RBP-specific activations in Figs. 4 and 5). In Section 2.4 (line 251), we moved the time indication closer towards the described phenomena of activity reversal, so that it is not confused with the time of the task switching, as in the previous version: “While in the other bands (the low-frequency bands and HGB) there was a reversal in the activity (around $t = 0.6$ s) ...”

Version 1:

Reviewer comments:

Reviewer #1

(Remarks to the Author)

The authors have addressed my comments and suggestion satisfactorily. Many congratulations on the work!

Reviewer #2

(Remarks to the Author)

I am happy with how the authors have addressed all my comments.

Reviewer #3

(Remarks to the Author)

I would like to thank the authors for their extensive replies and changes to the manuscript. I have no further comments.
